# LINK PREDICTION WITH NON-CONTRASTIVE LEARNING

**William Shiao**[1]*, **Zhichun Guo**[2]*, **Tong Zhao**[3],
**Evangelos E. Papalexakis**[1], **Yozen Liu**[3], **Neil Shah**[3]
[1]University of California, Riverside  [2]University of Notre Dame  [3]Snap Inc.
[1]{wshia002,epapalex}@ucr.edu, [2]zguo5@nd.edu, [3]{tzhao,yliu2,nshah}@snap.com

## ABSTRACT

Graph neural networks (GNNs) are prominent in the graph machine learning domain, owing to their strong performance across various tasks. A recent focal area is the space of graph self-supervised learning (SSL), which aims to derive useful node representations without labeled data. Notably, many state-of-the-art graph SSL approaches are *contrastive* methods, which use a combination of positive and negative samples to learn node representations. Owing to challenges in negative sampling (slowness and model sensitivity), recent literature introduced *non-contrastive* methods, which instead only use positive samples. Though such methods have shown promising performance in node-level tasks, their suitability for link prediction tasks, which are concerned with predicting link existence between pairs of nodes, and have broad applicability to recommendation systems contexts, is yet unexplored. In this work, we extensively evaluate the performance of existing non-contrastive methods for link prediction in both transductive and inductive settings. While most existing non-contrastive methods perform poorly overall, we find that, surprisingly, BGRL generally performs well in transductive settings. However, it performs poorly in the more realistic inductive settings where the model has to generalize to links to/from unseen nodes. We find that non-contrastive models tend to overfit to the training graph and use this analysis to propose T-BGRL, a novel non-contrastive framework that incorporates cheap corruptions to improve the generalization ability of the model. This simple modification strongly improves inductive performance in **5/6** of our datasets, with up to a **120%** improvement in Hits@50—all with comparable speed to other non-contrastive baselines, and up to **14×** faster than the best-performing contrastive baseline. Our work imparts interesting findings about non-contrastive learning for link prediction and paves the way for future researchers to further expand upon this area.

## 1 INTRODUCTION

Graph neural networks (GNNs) are ubiquitously used modeling tools for relational graph data, with widespread applications in chemistry (Chen et al., 2019; Guo et al., 2021; 2022a; Liu et al., 2022), forecasting and traffic prediction (Derrow-Pinion et al., 2021; Tang et al., 2020), recommendation systems (Ying et al., 2018b; He et al., 2020; Sankar et al., 2021; Tang et al., 2022; Fan et al., 2022), graph generation (You et al., 2018; Fan & Huang, 2019; Shiao & Papalexakis, 2021), and more. Given significant challenges in obtaining labeled data, one particularly exciting recent direction is the advent of graph self-supervised learning (SSL), which aims to learn representations useful for various downstream tasks without using explicit supervision besides available graph structure and node features (Zhu et al., 2020; Jin et al., 2021; Thakoor et al., 2022; Bielak et al., 2022).

One prominent class of graph SSL approaches are contrastive methods (Jin et al., 2020). These methods typically utilize contrastive losses such as InfoNCE (Oord et al., 2018) or margin-based losses (Ying et al., 2018b) between node and negative sample representations. However, such methods usually require either many negative samples (Hassani & Ahmadi, 2020) or carefully chosen ones (Ying et al., 2018b; Yang et al., 2020), where the first one results with quadratic number of

---

*Work done while interning at Snap Inc.

in-batch comparisons, and the latter is especially expensive on graphs since we often store the sparse adjacency matrix instead of its dense complement (Thakoor et al., 2022; Bielak et al., 2022). These drawbacks motivated the development of non-contrastive methods (Thakoor et al., 2022; Bielak et al., 2022; Zhang et al., 2021; Kefato & Girdzijauskas, 2021), based on advances in the image domain (Grill et al., 2020; Chen & He, 2021; Chen et al., 2020), which do not require negative samples and solely rely on augmentations. This allows for a large speedup compared to their contrastive counterparts with strong performance (Bielak et al., 2022; Zhang et al., 2021).

However, non-contrastive SSL methods are typically evaluated on node-level tasks, which is a more direct analog of image classification in the graph domain. In comparison, the link-level task (link prediction), which focuses on predicting link existence between pairs of nodes, is largely overlooked. This presents a critical gap in understanding: *Are non-contrastive methods suitable for link prediction tasks? When do they (not) work, and why?* This gap presents a huge opportunity, since link prediction is a cornerstone in the recommendation systems community (He et al., 2020; Zhang & Chen, 2019; Berg et al., 2017).

**Present Work.** To this end, our work first performs an extensive evaluation of non-contrastive SSL methods in link prediction contexts to discover the impact of different augmentations, architectures, and non-contrastive losses. We evaluate all of the (to the best of our knowledge) currently existing non-contrastive methods: CCA-SSG (Zhang et al., 2021), Graph Barlow Twins (GBT) (Bielak et al., 2022), and Bootstrapped Graph Latents (BGRL) (Thakoor et al., 2022) (which has the same design as the independently proposed SelfGNN (Kefato & Girdzijauskas, 2021)). We also compare these methods against a baseline end-to-end GCN (Kipf & Welling, 2017) with cross-entropy loss, and two contrastive baselines: GRACE (Zhu et al., 2020), and a GCN trained with max-margin loss (Ying et al., 2018a). We evaluate the methods in the transductive setting and find that BGRL (Thakoor et al., 2022) greatly outperforms not only the other non-contrastive methods, but also GRACE—a strong augmentation-based contrastive model for node classification. Surprisingly, BGRL even performs on-par with a margin-loss GCN (with the exception of 2/6 datasets). However, in the more realistic inductive setting, which considers prediction between new edges and nodes at inference time, we observe a huge gap in performance between BGRL and a margin-loss GCN (ML-GCN). Upon investigation, we find that BGRL is unable to sufficiently push apart the representations of negative links from positive links when new nodes are introduced, owing to a form of overfitting. To address this, we propose T-BGRL, a novel non-contrastive method which uses a corruption function to generate cheap "negative" samples—without performing the expensive negative sampling step of contrastive methods. We show that it greatly reduces overfitting tendencies, and outperforms existing non-contrastive methods across 5/6 datasets on the inductive setting. We also show that it maintains comparable speed with BGRL, and is 14× faster than the margin-loss GCN on the `Coauthor-Physics` dataset.

**Main Contributions**. In short, our main contributions are as follows:

- To the best of our knowledge, this is the first work to explore link prediction with non-contrastive SSL methods.

- We show that, perhaps surprisingly, BGRL (an existing non-contrastive model) works well in the transductive link prediction, with performance at par with contrastive baselines, implicitly behaving similarly to other contrastive models in pushing apart positive and negative node pairs.

- We show that non-contrastive SSL models underperform their contrastive counterparts in the inductive setting, and notice that they generalize poorly due to a lack of negative examples.

- Equipped with this understanding, we propose T-BGRL, a novel non-contrastive method that uses cheap "negative" samples to improve generalization. T-BGRL is simple to implement, very efficient when compared to contrastive methods, and improves on BGRL's inductive performance in 5/6 datasets, making it at or above par with the best contrastive baselines.

## 2 PRELIMINARIES

**Notation.** We denote a graph as $G = (\mathcal{V}, \mathcal{E})$, where $\mathcal{V}$ is the set of $n$ nodes (i.e., $n = |\mathcal{V}|$) and $\mathcal{E} \subseteq \mathcal{V} \times \mathcal{V}$ be the set of edges. Let the node-wise feature matrix be denoted by $\boldsymbol{X} \in \mathbb{R}^{n \times f}$, where $f$ is the number of raw features, and its $i$-th row $\boldsymbol{x}_i$ is the feature vector for the $i$-th node.

Let $\boldsymbol{A} \in \{0,1\}^{n \times n}$ denote the binary adjacency matrix. We denote the graph's learned node representations as $\boldsymbol{H} \in \mathbb{R}^{n \times d}$, where $d$ is the size of latent dimension, and $\boldsymbol{h}_i$ is the representation for the $i$-th node. Let $\boldsymbol{Y} \in \{0,1\}^{n \times n}$ be the desired output for link prediction, as $\mathcal{E}$ and $\boldsymbol{A}$ may have validation and test edges masked off. Similarly, let $\hat{\boldsymbol{Y}} \in \{0,1\}^{n \times n}$ be the output predicted by the decoder for link prediction. Let ORC be a perfect oracle function for our link prediction task, i.e., $\text{ORC}(\boldsymbol{A}, \boldsymbol{X}) = \boldsymbol{Y}$. Let $\text{NEIGH}(u) = \{v \mid (u,v) \in \mathcal{E} \vee (v,u) \in \mathcal{E}\}$. Note that we use the terms "embedding" and "representation" interchangeably in this work.

**GNNs for Link Prediction.** Many new approaches have also been developed with the recent advent of graph neural networks (GNNs). A predominant paradigm is the use of node-embedding-based methods (Hamilton et al., 2017; Berg et al., 2017; Ying et al., 2018b; Zhao et al., 2022b). Node-embedding-based methods typically consist of an encoder $\boldsymbol{H} = \text{ENC}(\boldsymbol{A}, \boldsymbol{X})$ and a decoder $\text{DEC}(\boldsymbol{H})$. The encoder model is typically a message-passing based Graph Neural Network (GNN) (Kipf & Welling, 2017; Hamilton et al., 2017; Zhang et al., 2020). The message-passing iterations of a GNN for a node $u$ can be described as follows:

$$\boldsymbol{h}_u^{(k+1)} = \text{UPDATE}^{(k)}\left(\boldsymbol{h}_u^{(k)}, \text{AGGREGATE}^{(k)}(\{\boldsymbol{h}_v^{(k)}, \forall v \in \text{NEIGH}(u)\})\right) \tag{1}$$

where UPDATE and AGGREGATE are differentiable functions, and $\boldsymbol{h}_u^{(0)} = \boldsymbol{x}_u$. The decoder model is usually an inner product or MLP applied on a concatenation of Hadamard product of the source and target learned node representations (Rendle et al., 2020; Wang et al., 2021).

**Graph SSL.** Below, we define a few terms used throughout our work which helps set the context for our discussion.

**Definition 2.1** (Augmentation). An augmentation $\text{AUG}^+$ is a label-preserving random transformation function $\text{AUG}^+ : (\boldsymbol{A}, \boldsymbol{X}) \rightarrow (\tilde{\boldsymbol{A}}, \tilde{\boldsymbol{X}})$ that does not change the oracle's expected value: $\mathbb{E}[\text{ORC}(\text{AUG}^+(\boldsymbol{A}, \boldsymbol{X}))] = \boldsymbol{Y}$.

**Definition 2.2** (Corruption). A corruption $\text{AUG}^-$ is a label-altering random transformation $\text{AUG}^- : (\boldsymbol{A}, \boldsymbol{X}) \rightarrow (\check{\boldsymbol{A}}, \check{\boldsymbol{X}})$ that changes the oracle's expected value: $\mathbb{E}[\text{ORC}(\text{AUG}^-(\boldsymbol{A}, \boldsymbol{X}))] \neq \boldsymbol{Y}$.[1]

**Definition 2.3** (Contrastive Learning). Contrastive methods select anchor samples (e.g. nodes) and then compare those samples to both *positive* samples (e.g. neighbors) and *negative* samples (e.g. non-neighbors) relative to those anchor samples.

**Definition 2.4** (Non-Contrastive Learning). Non-contrastive methods select anchor samples, but only compare those samples to variants of themselves, without leveraging other samples in the dataset.

**BGRL.** While we examine the performance of all of the non-contrastive graph models, we focus our detailed analysis exclusively on BGRL[2] (Thakoor et al., 2022) due to its superior performance in link prediction when compared to GBT (Bielak et al., 2022) and CCA-SSG (Zhang et al., 2021). BGRL consists of two encoders, one of which is referred to as the *online* encoder $\text{ENC}_\theta$; the other is referred to as the *target* encoder $\text{ENC}_\phi$. BGRL also incorporates a predictor PRED (typically a MLP) and two sets of augmentations: $\mathcal{A}_1^+, \mathcal{A}_2^+$. A single training step for BGRL is as follows: (a) we apply these augmentations: $(\tilde{\boldsymbol{A}}^{(1)}, \tilde{\boldsymbol{X}}^{(1)}) = \text{AUG}_1^+(\boldsymbol{A}, \boldsymbol{X})$; $(\tilde{\boldsymbol{A}}^{(2)}, \tilde{\boldsymbol{X}}^{(2)}) = \text{AUG}_2^+(\boldsymbol{A}, \boldsymbol{X})$. (b) we perform forward propagation $\boldsymbol{H} = \text{ENC}(\tilde{\boldsymbol{A}}^{(1)}, \tilde{\boldsymbol{X}}^{(1)})$; $\boldsymbol{H}_2 = \text{ENC}(\tilde{\boldsymbol{A}}^{(2)}, \tilde{\boldsymbol{X}}^{(2)})$. (c) we pass the output through the predictor $\boldsymbol{Z} = \text{PRED}(\boldsymbol{H}_1)$. (d) we use the mean pairwise cosine distance of $\boldsymbol{Z}$ and $\boldsymbol{H}_2$ as the loss (see Eqn. 2). (e) $\text{ENC}_\theta$ is updated via backpropagation and $\text{ENC}_\phi$ is updated via exponential moving average (EMA) from $\text{ENC}_\theta$. The BGRL loss is as follows:

$$\mathcal{L}_{\text{BGRL}} = -\frac{2}{n} \sum_{i=0}^{n-1} \frac{\tilde{\boldsymbol{z}}_i \cdot \boldsymbol{h}_i^{(2)}}{||\tilde{\boldsymbol{z}}_i|| \, ||\boldsymbol{h}_i^{(2)}||} \tag{2}$$

In the next section, we evaluate BGRL and other non-contrastive link prediction methods against contrastive baselines.

---

[1] Note that the definition of these functions are different from the corruption functions in Zhu et al. (2020) (which we define as *augmentations*) and are instead similar to the corruption functions in Veličković et al. (2018).

[2] Self-GNN (Kefato & Girdzijauskas, 2021), which was published independently, also shares the same architecture. As such, we refer to these two methods as BGRL.

## 3 DO NON-CONTRASTIVE LEARNING METHODS PERFORM WELL ON LINK PREDICTION TASKS?

Several non-contrastive methods have been proposed and have shown effectiveness in node classification (Kefato & Girdzijauskas, 2021; Thakoor et al., 2022; Zhang et al., 2021; Bielak et al., 2022). However, none of these methods evaluate or target link prediction tasks. We thus aim to answer the following questions: First, how well do these methods work for link prediction compared to existing contrastive/end-to-end baselines? Second, do they work equally well in both transductive and inductive settings? Finally, if they do work, why; if not, why not?

**Differences from Node Classification.** Link prediction differs from node classification in several key aspects. First, we must consider the embedding of both the source and destination nodes. Second, we have a much larger set of candidates for the same graph—$O(n^2)$ instead of $O(n)$. Finally, in real applications, link prediction is usually treated as a ranking problem, where we want positive links to be ranked higher than negative links, rather than as a classification problem, e.g. in recommendation systems, where we want to retrieve the top-$k$ most likely links (Cremonesi et al., 2010; Hubert et al., 2022). We discuss this in more detail in Section 3.1 below. Given these differences, it is unclear if methods performing well on node classification naturally perform well on link prediction tasks.

**Ideal Link Prediction.** What does it mean to perform well on link prediction? We clarify this point here. For some nodes $u, v, w \in \mathcal{V}$, let $(u, v) \in \mathcal{E}$ and $(u, w) \notin \mathcal{E}$. Then, an ideal encoder for link prediction would have $\text{DIST}(\boldsymbol{h}_u, \boldsymbol{h}_v) < \text{DIST}(\boldsymbol{h}_u, \boldsymbol{h}_w)$ for some distance function DIST. This idea is the core motivation behind margin-loss-based models (Ying et al., 2018a; Hamilton et al., 2017).

### 3.1 EVALUATION

**Datasets.** We use datasets from three different domains: citation networks, co-authorship networks, and co-purchase networks. We use the `Cora` and `Citeseer` citation networks (Sen et al., 2008), the `Coauthor-CS` and `Coauthor-Physics` co-authorship networks, and the `Amazon-Computers` and `Amazon-Photos` co-purchase networks (Shchur et al., 2018). We include dataset statistics in Appendix A.1.

**Metric.** Following work in the heterogeneous information network (Chen et al., 2018), knowledge-graph (Lin et al., 2015), and recommendation systems (Cremonesi et al., 2010; Hubert et al., 2022) communities, we choose to use Hits@$k$ over AUC-ROC metrics, since we often empirically prioritize ranking candidate links from a selected node context (e.g. ranking the probability that user $A$ will buy item $B$, $C$, or $D$), as opposed to arbitrarily ranking a randomly chosen positive over negative link (e.g. ranking whether the probability that user $A$ buys item $B$ is more likely than user $C$ does not buy item $D$). We report Hits@50 ($k = 50$) to strike a balance between the smaller datasets like `Cora` and the larger datasets like `Coauthor-Physics`. However, for completeness of the evaluation, we also include AUC-ROC results in Appendix A.8.

**Decoder.** Since our goal is to evaluate the performance of the encoder, we use the same decoder for all of our experiments across all of the methods. The choice of decoder has also been previously studied (Wang et al., 2021; 2022), so we use the best-performing decoder - a Hadamard product MLP. For a candidate link $(u, v)$, we have $\hat{\boldsymbol{Y}} = \text{DEC}(\boldsymbol{h}_u * \boldsymbol{h}_v)$ where $*$ represents the Hadamard product, and DEC is a two-layer MLP (with 256 hidden units) followed by a sigmoid. For the self-supervised methods, we first train the encoder and freeze its weights before training the decoder. As a contextual baseline, we also report results on an end-to-end GCN (E2E-GCN), for which we train the encoder and decoder jointly, backpropagating a binary cross-entropy loss on link existence.

#### 3.1.1 TRANSDUCTIVE EVALUATION

**Transductive Setting.** We first evaluate the performance of the methods in the transductive setting, where we train on $G_{train} = (\mathcal{V}, \mathcal{E}_{train})$ for $\mathcal{E}_{train} \subset \mathcal{E}$, validate our method on $G_{val} = (\mathcal{V}, \mathcal{E}_{val})$ for $\mathcal{E}_{val} \subset (\mathcal{E} - \mathcal{E}_{train})$, and test on $G_{test} = (\mathcal{V}, \mathcal{E}_{test})$ for $\mathcal{E}_{test} = \mathcal{E} - \mathcal{E}_{train} - \mathcal{E}_{val}$. Note that the same nodes are present in training, validation, and testing. We also do not introduce any new edges during inference time—inference is performed on $\mathcal{E}_{train}$.

**Results.** The results of our evaluation are shown in Table 1. As expected, the end-to-end GCN generally performs the best across all of the datasets. We also find that CCA-SSG and GBT similarly

Table 1: Transductive performance of different link prediction methods. We **bold** the best-performing method and underline the second-best method for each dataset. BGRL consistently outperforms other non-contrastive methods and GRACE, and also outperforms ML-GCN, on 3/6 datasets.

| Dataset | End-To-End | Contrastive | | Non-Contrastive | | |
|---|---|---|---|---|---|---|
| | E2E-GCN | ML-GCN | GRACE | CCA-SSG | GBT | BGRL |
| Cora | **0.816**$_{\pm0.013}$ | 0.815$_{\pm0.002}$ | 0.686$_{\pm0.056}$ | 0.348$_{\pm0.091}$ | 0.460$_{\pm0.149}$ | 0.792$_{\pm0.015}$ |
| Citeseer | 0.822$_{\pm0.017}$ | 0.771$_{\pm0.020}$ | 0.707$_{\pm0.068}$ | 0.249$_{\pm0.168}$ | 0.472$_{\pm0.196}$ | **0.858**$_{\pm0.020}$ |
| Amazon-Photos | **0.642**$_{\pm0.029}$ | 0.430$_{\pm0.032}$ | 0.486$_{\pm0.025}$ | 0.369$_{\pm0.013}$ | 0.434$_{\pm0.038}$ | 0.562$_{\pm0.013}$ |
| Amazon-Computers | **0.426**$_{\pm0.036}$ | 0.320$_{\pm0.060}$ | 0.240$_{\pm0.027}$ | 0.201$_{\pm0.032}$ | 0.258$_{\pm0.008}$ | 0.346$_{\pm0.018}$ |
| Coauthor-CS | 0.762$_{\pm0.010}$ | **0.787**$_{\pm0.011}$ | 0.456$_{\pm0.066}$ | 0.229$_{\pm0.018}$ | 0.298$_{\pm0.033}$ | 0.515$_{\pm0.016}$ |
| Coauthor-Physics | 0.798$_{\pm0.018}$ | **0.810**$_{\pm0.003}$ | OOM | 0.157$_{\pm0.009}$ | 0.187$_{\pm0.011}$ | 0.476$_{\pm0.015}$ |

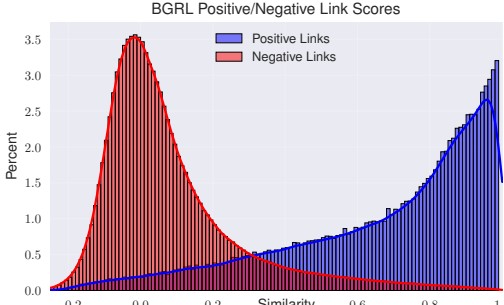 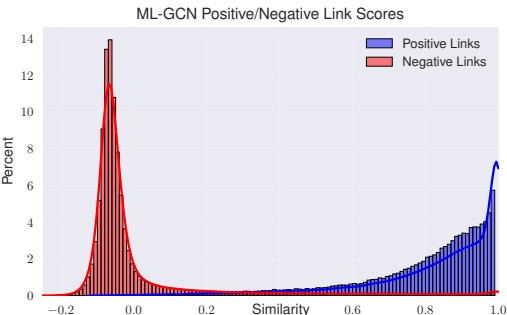

Figure 1: These plots show similarities between node embeddings. **Left:** distribution of positive/negative link similarities for BGRL. **Right:** distribution of positive/negative link similarities for ML-GCN. We can see that while they behave similarly, the ML-GCN does a better job of ensuring that positive/negative links are well separated. These scores are computed on `Amazon-Photos`.

perform poorly relative to the other methods. This is intuitive, as neither method was designed for link prediction and were only evaluated for node classification in their respective papers. Surprisingly, however, BGRL outperforms the ML-GCN (the strongest contrastive baseline) on 3/6 of the datasets and performs similarly on 1 other (`Cora`). It also outperforms GRACE across all of the datasets.

**Understanding BGRL Performance.** Interestingly, we find that BGRL exhibits similar behavior to the ML-GCN on many datasets, despite the BGRL loss function (see Equation (2)) not explicitly optimizing for this. Relative to an anchor node $u$, we can express the max-margin loss of the ML-GCN as follows:

$$L(u) = \mathbb{E}_{v \sim \text{NEIGH}(u)} \left[ \mathbb{E}_{w \sim \mathcal{E} - \text{NEIGH}(u)} J(u, v, w) \right] \tag{3}$$

where $J(u, v, w)$ is the margin ranking loss for an anchor $u$, positive sample $v$, and negative $w$:

$$J(u, v, w) = \max\{0, \boldsymbol{h}_u \cdot \boldsymbol{h}_v - \boldsymbol{h}_u \cdot \boldsymbol{h}_w + \Delta\} \tag{4}$$

and $\Delta$ is a hyperparameter for the size of the margin. This seemingly explicitly optimizes for the aforementioned ideal link prediction behavior (anchor-aware ranking of positive over negative links). Despite these circumstances, Figure 1 shows that both BGRL and ML-GCN both clearly separate positive and negative samples, although ML-GCN pushes them further apart. We provide some intuition on why this may occur in Appendix A.10 below.

**Why Does BGRL Not Collapse?** The loss function for BGRL (see Equation (2)) is 0 when $h_i^{(2)} = 0$ or $\tilde{z}_i = 0$, i.e., the loss is minimized when the model produces all-zero outputs. While theoretically possible, this is clearly undesirable behavior since this does not result in useful embeddings. We refer to this case as model collapse. It is not fully understood why non-contrastive models do not collapse, but there have been several reasons proposed in the image domain with both theoretical and empirical grounding. We discuss this more in Appendix A.9. Consistent with the findings from Thakoor et al. (2022), we find that collapse does not occur in practice (with reasonable hyperparameter selection).

**Conclusion.** We find that CCA-SSG and GBT generally perform poorly compared to contrastive baselines. Surprisingly, we find that BGRL generally performs well in the transductive setting by successfully separating positive and negative link distance distributions. However, this setting may

Table 2: Performance of various methods in the inductive setting. See Section 3.1.2 for an explanation of our inductive setting. Although we do not introduce T-BGRL until Section 4, we include the results here to save space.

| Dataset | End-To-End | Contrastive | | Non-Contrastive | | | |
| | E2E-GCN | ML-GCN | GRACE | GBT | CCA-SSG | BGRL | T-BGRL |
|---|---|---|---|---|---|---|---|
| Overall | | | | | | | |
| Cora | $\underline{0.523}_{\pm0.019}$ | $0.490_{\pm0.028}$ | $0.448_{\pm0.043}$ | $0.135_{\pm0.077}$ | $0.120_{\pm0.018}$ | $0.324_{\pm0.184}$ | $\mathbf{0.568}_{\pm0.033}$ |
| Citeseer | $\underline{0.621}_{\pm0.034}$ | $0.661_{\pm0.036}$ | $0.514_{\pm0.053}$ | $0.305_{\pm0.026}$ | $0.170_{\pm0.071}$ | $0.526_{\pm0.055}$ | $\mathbf{0.727}_{\pm0.027}$ |
| Coauthor-Cs | $0.484_{\pm0.048}$ | $\mathbf{0.572}_{\pm0.037}$ | $0.313_{\pm0.017}$ | $0.182_{\pm0.025}$ | $0.176_{\pm0.013}$ | $0.438_{\pm0.025}$ | $\underline{0.534}_{\pm0.026}$ |
| Coauthor-Physics | $0.386_{\pm0.016}$ | $\mathbf{0.550}_{\pm0.059}$ | OOM | $0.112_{\pm0.014}$ | $0.037_{\pm0.051}$ | $0.439_{\pm0.013}$ | $\underline{0.463}_{\pm0.023}$ |
| Amazon-Computers | $0.179_{\pm0.010}$ | $\underline{0.279}_{\pm0.044}$ | $0.212_{\pm0.057}$ | $0.172_{\pm0.015}$ | $0.155_{\pm0.013}$ | $0.270_{\pm0.034}$ | $\mathbf{0.312}_{\pm0.027}$ |
| Amazon-Photos | $0.420_{\pm0.123}$ | $\mathbf{0.478}_{\pm0.008}$ | $0.262_{\pm0.010}$ | $0.289_{\pm0.032}$ | $0.182_{\pm0.072}$ | $\underline{0.460}_{\pm0.023}$ | $0.450_{\pm0.017}$ |
| Performance on Observed-Observed Node Edges | | | | | | | |
| Cora | $\underline{0.574}_{\pm0.020}$ | $0.490_{\pm0.029}$ | $0.557_{\pm0.038}$ | $0.149_{\pm0.084}$ | $0.124_{\pm0.026}$ | $0.345_{\pm0.196}$ | $\mathbf{0.624}_{\pm0.027}$ |
| Citeseer | $0.610_{\pm0.023}$ | $\underline{0.621}_{\pm0.021}$ | $0.602_{\pm0.050}$ | $0.358_{\pm0.031}$ | $0.197_{\pm0.082}$ | $0.605_{\pm0.045}$ | $\mathbf{0.768}_{\pm0.021}$ |
| Coauthor-Cs | $0.504_{\pm0.047}$ | $\mathbf{0.591}_{\pm0.034}$ | $0.332_{\pm0.018}$ | $0.187_{\pm0.023}$ | $0.177_{\pm0.013}$ | $0.462_{\pm0.025}$ | $\underline{0.535}_{\pm0.026}$ |
| Coauthor-Physics | $0.390_{\pm0.015}$ | $\mathbf{0.566}_{\pm0.058}$ | OOM | $0.117_{\pm0.014}$ | $0.039_{\pm0.054}$ | $0.445_{\pm0.012}$ | $\underline{0.469}_{\pm0.023}$ |
| Amazon-Computers | $0.177_{\pm0.009}$ | $\underline{0.278}_{\pm0.044}$ | $0.212_{\pm0.059}$ | $0.169_{\pm0.016}$ | $0.155_{\pm0.014}$ | $0.270_{\pm0.034}$ | $\mathbf{0.313}_{\pm0.027}$ |
| Amazon-Photos | $0.418_{\pm0.123}$ | $\mathbf{0.483}_{\pm0.009}$ | $0.265_{\pm0.011}$ | $0.295_{\pm0.031}$ | $0.185_{\pm0.070}$ | $\underline{0.467}_{\pm0.023}$ | $0.457_{\pm0.015}$ |
| Performance on Observed-Unobserved Node Edges | | | | | | | |
| Cora | $0.462_{\pm0.023}$ | $\underline{0.487}_{\pm0.021}$ | $0.367_{\pm0.045}$ | $0.128_{\pm0.075}$ | $0.115_{\pm0.014}$ | $0.309_{\pm0.175}$ | $\mathbf{0.528}_{\pm0.037}$ |
| Citeseer | $0.645_{\pm0.055}$ | $\underline{0.705}_{\pm0.039}$ | $0.458_{\pm0.063}$ | $0.280_{\pm0.024}$ | $0.148_{\pm0.067}$ | $0.487_{\pm0.064}$ | $\mathbf{0.708}_{\pm0.034}$ |
| Coauthor-Cs | $0.459_{\pm0.049}$ | $\mathbf{0.545}_{\pm0.042}$ | $0.284_{\pm0.017}$ | $0.175_{\pm0.026}$ | $0.177_{\pm0.013}$ | $0.402_{\pm0.025}$ | $\underline{0.536}_{\pm0.027}$ |
| Coauthor-Physics | $0.379_{\pm0.019}$ | $\mathbf{0.525}_{\pm0.058}$ | OOM | $0.106_{\pm0.013}$ | $0.035_{\pm0.048}$ | $0.429_{\pm0.013}$ | $\underline{0.455}_{\pm0.022}$ |
| Amazon-Computers | $0.183_{\pm0.010}$ | $\underline{0.281}_{\pm0.045}$ | $0.213_{\pm0.056}$ | $0.177_{\pm0.014}$ | $0.155_{\pm0.011}$ | $0.270_{\pm0.034}$ | $\mathbf{0.312}_{\pm0.027}$ |
| Amazon-Photos | $0.424_{\pm0.123}$ | $\mathbf{0.470}_{\pm0.007}$ | $0.258_{\pm0.011}$ | $0.279_{\pm0.032}$ | $0.178_{\pm0.076}$ | $\underline{0.449}_{\pm0.022}$ | $0.439_{\pm0.021}$ |
| Performance on Unobserved-Unobserved Node Edges | | | | | | | |
| Cora | $0.239_{\pm0.027}$ | $\mathbf{0.507}_{\pm0.063}$ | $0.252_{\pm0.066}$ | $0.100_{\pm0.076}$ | $0.125_{\pm0.020}$ | $0.287_{\pm0.164}$ | $\underline{0.463}_{\pm0.065}$ |
| Citeseer | $0.595_{\pm0.073}$ | $\mathbf{0.681}_{\pm0.101}$ | $0.287_{\pm0.039}$ | $0.137_{\pm0.019}$ | $0.126_{\pm0.043}$ | $0.271_{\pm0.078}$ | $\underline{0.595}_{\pm0.045}$ |
| Coauthor-Cs | $0.372_{\pm0.043}$ | $\underline{0.483}_{\pm0.046}$ | $0.230_{\pm0.019}$ | $0.159_{\pm0.037}$ | $0.157_{\pm0.011}$ | $0.341_{\pm0.032}$ | $\mathbf{0.517}_{\pm0.032}$ |
| Coauthor-Physics | $0.365_{\pm0.024}$ | $\mathbf{0.505}_{\pm0.065}$ | OOM | $0.098_{\pm0.013}$ | $0.034_{\pm0.047}$ | $0.424_{\pm0.014}$ | $\underline{0.445}_{\pm0.026}$ |
| Amazon-Computers | $0.183_{\pm0.008}$ | $\underline{0.275}_{\pm0.046}$ | $0.214_{\pm0.052}$ | $0.181_{\pm0.015}$ | $0.155_{\pm0.012}$ | $0.265_{\pm0.032}$ | $\mathbf{0.305}_{\pm0.029}$ |
| Amazon-Photos | $0.419_{\pm0.126}$ | $\mathbf{0.461}_{\pm0.014}$ | $0.251_{\pm0.010}$ | $0.265_{\pm0.044}$ | $0.172_{\pm0.084}$ | $\underline{0.442}_{\pm0.028}$ | $0.416_{\pm0.027}$ |

not be representative of real-world problems. In the next section, we evaluate the methods in the more realistic inductive setting to see if this performance holds.

### 3.1.2 Inductive Evaluation

**Inductive Setting.** While we observe some promising results in favor of non-contrastive methods (namely, BGRL) in the transductive setting, we note that this setting is not entirely realistic. In practice, we often have both new nodes and edges introduced at inference time after our model is trained. For example, consider a social network upon which a model is trained at some time $t_1$ but is used for inference (for a GNN, this refers to the message-passing step) at time $t_2$, where new users and friendships have been added to the network in the interim. Then, the goal of a model run at time $t_2$ would be to predict any new links at new network state $t_3$ (although we assume there are no new nodes introduced at that step since we cannot compute the embedding of nodes without performing inference on them first). To simulate this setting, we first partition the graph into two sets of nodes: "observed" nodes (that we see during training) and "unobserved nodes" (that are only used for inference and testing). We then withhold a portion of the edges at each of the time steps $t_3, t_2, t_1$ to serve as testing-only, inference-only, and training-only edges, respectively. We describe this process in more detail in Appendix A.4.

**Results.** Table 2 shows that in the inductive setting, BGRL is outperformed by the contrastive ML-GCN on *all* datasets. It still outperforms CCA-SSG and GBT, but it is *much* less competitive in the inductive setting. We next ask: what accounts for this large difference in performance?

**Why Does BGRL Not Work Well in the Inductive Setting?** One possible reason for the poor performance of BGRL in the inductive setting is that it is unable to correctly differentiate unseen positives from unseen negatives, i.e., it is overfitting on the training graph. Intuitively, this could happen due to a lack of negative samples—BGRL never pushes samples away from each other. We show that this is indeed the case in Figure 2, where BGRL's negative link score distribution has heavy overlap with its positive link score distribution. We can also see this behavior in Figure 1 where the

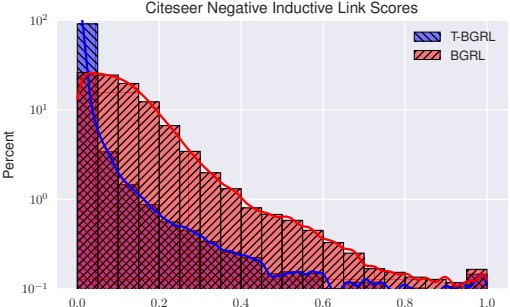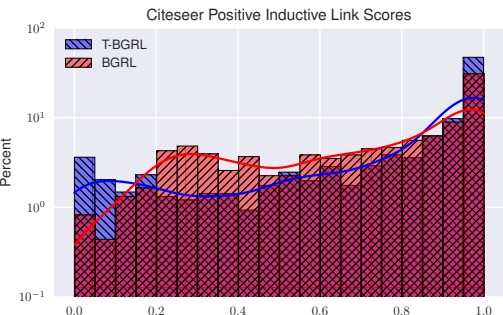

Figure 2: These plots show similarities between node embeddings on `Citeseer`. **Left:** distribution of similarity to *non-neighbors* for T-BGRL and BGRL. **Right:** distribution of similarity to *neighbors* for T-BGRL and BGRL. Note that the y-axis is on a logarithmic scale. T-BGRL clearly does a better job of ensuring that negative link representations are pushed far apart from those of positive links.

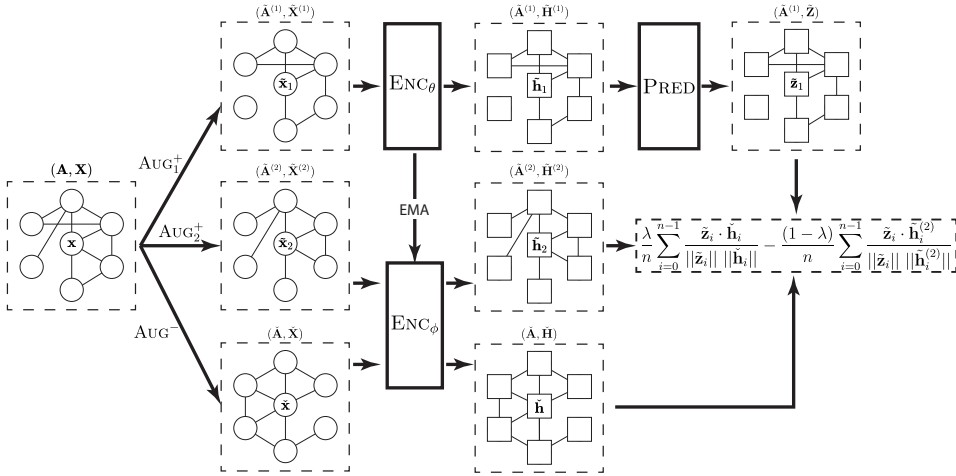

Figure 3: T-BGRL architecture diagram. The loss function is also shown in Equation (5).

ML-GCN does a clearly better job of pushing positive/negative samples far apart, despite BGRL's surprising success. Naturally, improving the separation between these distributions increases the chance of a correct prediction. We investigate this hypothesis in Section 4 below and propose T-BGRL (Figure 3), a novel method to help alleviate this issue.

## 4 IMPROVING INDUCTIVE PERFORMANCE IN A NON-CONTRASTIVE FRAMEWORK

In order to reduce this systematic gap in performance between ML-GCN (the best-performing contrastive model) and BGRL (the best-performing non-contrastive model), we observe that we need to push negative and positive node pair representations further apart. This way, pairs between new nodes—introduced at inference time—have a higher chance of being classified correctly. Contrastive methods utilize negative sampling for this purpose, but we wish to avoid negative sampling owing to high computational cost. In lieu of this, we propose a simple, yet powerfully effective idea below.

**Model Intuition.** To emulate the effect of negative sampling without actually performing it, we propose Triplet-BGRL (T-BGRL). In addition to the two augmentations performed during standard non-contrastive SSL training, we add a corruption to function as a cheap negative sample. For each node, like BGRL, we minimize the distance between its representations across two augmentations. However, taking inspiration from triplet-style losses (Hoffer & Ailon, 2014), we also maximize the distance between the augmentation and corruption representations.

**Model Design.** Ideally, this model should not only perform better than BGRL in the inductive setting, but should also have the same time complexity as BGRL. In order to meet these expectations, we design efficient, linear-time corruptions (same asymptotic runtime as the augmentations). We also choose to use the online encoder $\text{ENC}_\phi$ to generate embeddings for the corrupted graph so that

T-BGRL does not have any additional parameters. Figure 3 illustrates the overall architecture of the proposed T-BGRL, and Algorithm 1 presents PyTorch-style pseudocode. Our new proposed loss function is as follows:

$$\mathcal{L}_{\text{T-BGRL}} = \underbrace{\frac{\lambda}{n} \sum_{i=0}^{n-1} \frac{\tilde{z}_i \cdot \check{h}_i}{||\tilde{z}_i|| \, ||\check{h}_i||}}_{\text{T-BGRL Loss Term}} - \underbrace{\frac{(1-\lambda)}{n} \sum_{i=0}^{n-1} \frac{\tilde{z}_i \cdot h_i^{(2)}}{||\tilde{z}_i|| \, ||h_i^{(2)}||}}_{\text{BGRL Loss}} \tag{5}$$

where $\lambda$ is a hyperparameter controlling the repulsive forces between augmentation and corruption.

**Corruption Choice.** We experiment with several different corruptions methods, but limit ourselves to linear-time corruptions in order to maintain the efficiency of BGRL. We find that SHUFFLEFEATRANDOMEDGE$(A, X) = (\check{A}, \check{X})$, where $\check{A} \sim \{0,1\}^{n \times n}$ and $\check{X} =$ SHUFFLEROWS$(X)$ works the best. We describe each of the different corruptions we experimented with in Appendix A.7.

**Inductive Results.** Table 2 shows that T-BGRL improves inductive performance over BGRL in 5/6 datasets, with very large improvements in the `Cora` and `Citeseer` datasets. The only dataset where BGRL outperformed T-BGRL is the `Amazon-Photos` dataset. However, this gap is much smaller (0.01 difference in Hits@50) than the improvements on the other datasets. We plot the scores output by the decoder for unseen negative pairs compared to those for unseen positive pairs in Figure 2. We can see that T-BGRL pushes apart unseen negative and positive pairs much better than BGRL.

---

**Algorithm 1:** PyTorch-style pseudocode for T-BGRL

```
# Enc_o: online encoder network
# Enc_t: target encoder network
# Pred: predictor network
# lam: trade-off
# decay: EMA decay parameter
# g: input graph
# feat: node features

g1, feat1 = augment(g, feat) # augmentation #1
g2, feat2 = augment(g, feat) # augmentation #2
c_g, c_feat = corrupt(g, feat) # corruption

h1 = Enc_o(g1, feat1)
h2 = Enc_t(g2, feat2)
c_z = Enc_t(c_g, c_feat)
z1 = Pred(z1)

loss = lam*cosine_similarity(z1, c_z) \
       - (1-lam)*cosine_similarity(z1, h2)
loss.backward() # backprop

# Update Enc_t with EMA
Enc_t.params = decay * Enc_t.params \
               + (1-decay) * Enc_o.params
```

---

**Transductive Results.** We also evaluate the performance of T-BGRL in the transductive setting to ensure that it does not significantly reduce performance when compared to BGRL. See Table 3 on the right for the results.

**Difference from Contrastive Methods.** While our method shares some similarities with contrastive methods, we believe T-BGRL is strictly non-contrastive because it does not require the $O(n^2)$ sampling from the complement of the

Table 3: Transductive performance of T-BGRL compared to ML-GCN and BGRL (same numbers as Table 1 above; full figure in Table 5).

| Dataset | ML-GCN | BGRL | T-BGRL |
|---|---|---|---|
| `Cora` | **0.815** | 0.792 | $0.773_{\pm 0.020}$ |
| `Citeseer` | 0.771 | 0.858 | $\mathbf{0.868}_{\pm 0.023}$ |
| `Coauthor-Cs` | **0.787** | 0.515 | $0.555_{\pm 0.009}$ |
| `Coauthor-Physics` | **0.810** | 0.476 | $0.471_{\pm 0.021}$ |
| `Amazon-Computers` | 0.320 | **0.346** | $0.315_{\pm 0.015}$ |
| `Amazon-Photos` | 0.430 | **0.562** | $0.517_{\pm 0.016}$ |

edge index used by contrastive methods. This is clearly shown in Figure 4, where T-BGRL and BGRL have similar runtimes and are much faster than GRACE and ML-GCN. The corruption can be viewed as a "negative" augmentation—with the only difference being that it changes the expected label for each link. In fact, one of the corruptions that we consider, SPARSIFYFEATSPARSIFYEDGE, is essentially the same as the augmentations using by BGRL (except with much higher drop probability). We discuss other corruptions below in Appendix A.7.

**Scalability.** We evaluate the runtime of our model on different datasets. Figure 4 shows the running times to fully train a model for different contrastive and non-contrastive methods over 5 different runs. Note that we use a fixed 10,000 epochs for GRACE, CCA-SSG, GBT, BGRL, and T-BGRL, but use early stopping on the ML-GCN with a maximum of 1,000 epochs. We find that (i) T-BGRL is comparable to BGRL in runtime owing to efficient choices of corruptions, (ii) it is about $4.3\times$ faster than GRACE on `Amazon-Computers` (the largest dataset which GRACE can run on), and (ii) it is $14\times$ faster than ML-GCN. CCA-SSG is the fastest of all the methods but performs the worst. As mentioned above, we do not compare with SEAL (Zhang & Chen, 2018) or other subgraph-based methods due to how slow they are during inference. SUREL (Yin et al., 2022) is ~250× slower, and SEAL (Zhang & Chen, 2018) is about ~3900× slower according to Yin et al. (2022). In conclusion,

we find that T-BGRL is roughly as scalable as other non-contrastive methods, and much more scalable than existing contrastive methods.

## 5 OTHER RELATED WORK

**Link Prediction.** Link prediction is a long-standing graph machine learning task. Some traditional methods include (i) matrix (Menon & Elkan, 2011; Wang et al., 2020) or tensor factorization (Acar et al., 2009; Dunlavy et al., 2011) methods which factor the adjacency and/or feature matrices to derive node representations which can predict links equipped with inner products, and (ii) heuristic methods which score node pairs based on neighborhood and overlap (Yu et al., 2017; Zareie & Sakellariou, 2020; Philip et al., 2010). Several shallow graph embedding methods (Grover & Leskovec, 2016; Perozzi et al., 2014) which train node embeddings by random-walk strategies have also been used for link prediction. In addition to the node-

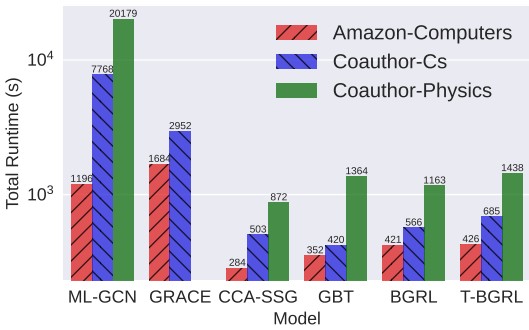

Figure 4: Total runtime comparison of different contrastive and non-contrastive methods. T-BGRL and BGRL have relatively similar runtimes and are significantly faster than the contrastive methods (GRACE and ML-GCN).

embedding-based GNN methods mentioned in Section 2, several works (Yin et al., 2022; Zhang & Chen, 2018; Hao et al., 2020) propose subgraph-based methods for this task, which aim to classify subgraphs around each candidate link. Few works focus on scalable link prediction with distillation (Guo et al., 2022b), decoder (Wang et al., 2022), and sketching designs (Chamberlain et al., 2022).

**Graph SSL Methods.** Most graph SSL methods can be put categorized into contrastive and non-contrastive methods. Contrastive learning has been applied to link prediction with margin-loss-based methods such as PinSAGE (Ying et al., 2018a), and GraphSAGE (Hamilton et al., 2017), where negative sample representations are pushed apart from positive sample representations. GRACE (Zhu et al., 2020) uses augmentation (Zhao et al., 2022a) during this negative sampling process to further increase the performance of the model. DGI (Veličković et al., 2018) leverages mutual information maximization between local patch and global graph representations. Some works (Ju et al., 2022; Jin et al., 2021) also explore using multiple contrastive pretext tasks for SSL. Several works (You et al., 2020; Lin et al., 2022) also focus on graph-level contrastive learning, via graph-level augmentations and careful negative selection. Recently, non-contrastive methods have been applied to graph representation learning. Self-GNN (Kefato & Girdzijauskas, 2021) and BGRL (Thakoor et al., 2022) use ideas from BYOL (Grill et al., 2020) and SimSiam (Chen & He, 2021) to propose a graph framework that does not require negative sampling. We describe BGRL in depth in Section 2 above. Graph Barlow Twins (GBT) (Bielak et al., 2022) is adapted from the Barlow Twins model in the image domain (Zbontar et al., 2021) and uses cross-correlation to learn node representations with a shared encoder. CCA-SSG (Zhang et al., 2021) uses ideas from Canonical Correlation Analysis (CCA) (Hotelling, 1992) and Deep CCA (Andrew et al., 2013) for their loss function. These models are somewhat similar in that it has also been shown that Barlow Twins is equivalent to Kernel CCA (Balestriero & LeCun, 2022).

## 6 CONCLUSION

To our knowledge, this is the first work to study non-contrastive SSL methods and their performance on link prediction. We first evaluate several contrastive and non-contrastive graph SSL methods on link prediction tasks, and find that surprisingly, one popular non-contrastive method (BGRL) is able to perform well in the transductive setting. We also observe that BGRL struggles in the inductive setting, and identify that it has a tendency to overfit the training graph, indicating it fails to push positive and negative node pair representations far apart from each other. Armed with these insights, we propose T-BGRL, a simple but effective non-contrastive strategy which works by generating extremely cheap "negatives" by corrupting the original inputs. T-BGRL sidesteps the expensive negative sampling step evident in contrastive learning, while enjoying strong performance benefits. T-BGRL improves on BGRL's inductive performance in 5/6 datasets while achieving similar transductive performance, making it comparable to the best contrastive baselines, but with a 14× speedup over the best contrastive methods.

## REPRODUCIBILITY STATEMENT

To ensure reproducibility, our source code is available online at https://github.com/snap-research/non-contrastive-link-prediction. The hyperparameters and instructions for reproducing all experiments are provided in the README.md file.

## ACKNOWLEDGMENTS

UCR coauthors were partly supported by the National Science Foundation under CAREER grant no. IIS 2046086 and were also sponsored by the Combat Capabilities Development Command Army Research Laboratory under Cooperative Agreement Number W911NF-13-2-0045 (ARL Cyber Security CRA). Any opinions, findings, and conclusions or recommendations expressed in this material are those of the author(s) and do not necessarily reflect the views of the funding parties.

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

# A APPENDIX

## A.1 DATASET STATISTICS

Table 4: Statistics for the datasets used in our work.

| Dataset | Nodes | Edges | Features |
|---------|-------|-------|----------|
| Cora | 2,708 | 5,278 | 1,433 |
| Citeseer | 3,327 | 4,552 | 3,703 |
| Coauthor-Cs | 18,333 | 163,788 | 6,805 |
| Coauthor-Physics | 34,493 | 495,924 | 8,415 |
| Amazon-Computers | 13,752 | 491,722 | 767 |
| Amazon-Photos | 7,650 | 238,162 | 745 |

## A.2 MACHINE DETAILS

We run all of our experiments on either NVIDIA P100 or V100 GPUs. We use machines with 12 virtual CPU cores and 24 GB of RAM for the majority of our experiments. We exclusively use V100s for our timing experiments. We ran our experiments on Google Cloud Platform.

## A.3 TRANSDUCTIVE SETTING DETAILS

We use an 85/5/10 split for training/validation/testing data—following Zhang & Chen (2018); Cai et al. (2020).

## A.4 INDUCTIVE SETTING DETAILS

The inductive setting represents a more realistic setting than the transductive setting. For example, consider a social network upon which a model is trained at some time $t_1$ but is used for inference (for a GNN, this refers to the message-passing step) at time $t_2$, where new users and friendships have been added to the network in the interim. Then, the goal of a model run at time $t_2$ would be to predict any new links at new network state $t_3$ (although we assume there are no new nodes introduced at that step since we cannot compute the embedding of nodes without performing inference on them first). To simulate this setting, we first perform the following steps:

1. We withhold a portion of the edges (and the same number of disconnected node pairs) to use as testing-only edges.
2. We partition the graph into two sets of nodes: "observed" nodes (that we see during training) and "unobserved nodes" (that can only be seen during testing).
3. We mask out some edges to use as testing-only edges.
4. We mask out some edges to use as inference-only edges.
5. We mask out some edges to use as validation-only edges.
6. We mask out some edges to use as training-only edges.

As the test edges are sampled before the node split, there will be three kinds of them after the splitting. Specifically: edges within observed nodes, edges between observed nodes and unobserved nodes, and edges within unobserved nodes. For ease of data preparation, we use the same percentages for the test edge splitting, unobserved node splitting, and validation edge splitting. Specifically, we mask out 30% of the edges (at each of the above stages) on the small datasets (Cora and Citeseer), and 10% on all the other datasets. We use a 30% split on the small datasets to ensure that we have a sufficient number of edges for testing and validation purposes.

## A.5 EXPERIMENTAL SETUP

To ensure that we fairly evaluate each model, we run a Bayesian hyperparameter sweep for 25 runs across each model-dataset combination with the target metric being the validation Hits@50. Each run

Table 6: Area under the ROC curve for the methods in the transductive setting.

| Dataset | End-To-End | Contrastive | | Non-Contrastive | | | |
|---|---|---|---|---|---|---|---|
| | E2E-GCN | ML-GCN | GRACE | CCA-SSG | GBT | BGRL | T-BGRL |
| Cora | $\mathbf{0.911}_{\pm 0.004}$ | $0.893_{\pm 0.007}$ | $0.883_{\pm 0.020}$ | $0.647_{\pm 0.076}$ | $0.736_{\pm 0.109}$ | $\mathbf{0.911}_{\pm 0.008}$ | $\underline{0.910}_{\pm 0.005}$ |
| Citeseer | $0.922_{\pm 0.006}$ | $0.891_{\pm 0.006}$ | $0.863_{\pm 0.042}$ | $0.661_{\pm 0.050}$ | $0.755_{\pm 0.120}$ | $\underline{0.934}_{\pm 0.009}$ | $\mathbf{0.953}_{\pm 0.003}$ |
| Coauthor-Cs | $\underline{0.964}_{\pm 0.005}$ | $\mathbf{0.966}_{\pm 0.001}$ | $0.961_{\pm 0.003}$ | $0.758_{\pm 0.047}$ | $0.894_{\pm 0.017}$ | $0.959_{\pm 0.002}$ | $0.956_{\pm 0.002}$ |
| Coauthor-Physics | $\underline{0.978}_{\pm 0.001}$ | $\mathbf{0.986}_{\pm 0.000}$ | OOM | $0.821_{\pm 0.051}$ | $0.834_{\pm 0.084}$ | $0.961_{\pm 0.002}$ | $0.963_{\pm 0.001}$ |
| Amazon-Computers | $\mathbf{0.985}_{\pm 0.001}$ | $\underline{0.983}_{\pm 0.001}$ | $0.951_{\pm 0.011}$ | $0.907_{\pm 0.025}$ | $0.946_{\pm 0.007}$ | $0.969_{\pm 0.002}$ | $0.976_{\pm 0.001}$ |
| Amazon-Photos | $\mathbf{0.989}_{\pm 0.000}$ | $\underline{0.983}_{\pm 0.002}$ | $0.981_{\pm 0.001}$ | $0.939_{\pm 0.008}$ | $0.956_{\pm 0.015}$ | $0.980_{\pm 0.000}$ | $0.982_{\pm 0.000}$ |

is the result of the mean averaged over 5 runs (retraining both the encoder and decoder). We used the Weights and Biases (Biewald, 2020) Bayesian optimizer for our experiments. We provide a sample configuration file to reproduce our sweeps, as well as the exact parameters used for the top T-BGRL runs shown in our tables.

We used the reference GRACE implementation and BGRL implementation but modified them for link prediction instead of node classification. We based our E2E-GCN off of the OGB (Hu et al., 2020) implementation. We re-implemented CCA-SSG and GBT. The code for all of our implementations and modifications can be found in the link in our paper above.

## A.6 FULL RESULTS

Table 5 shows the results of all the methods (including T-BGRL) on transductive setting.

Table 5: Full transductive performance table (combination of Tables 1 and 3).

| Dataset | End-To-End | Contrastive | | Non-Contrastive | | | |
|---|---|---|---|---|---|---|---|
| | E2E-GCN | ML-GCN | GRACE | CCA-SSG | GBT | BGRL | T-BGRL |
| Cora | $\mathbf{0.816}_{\pm 0.013}$ | $\underline{0.815}_{\pm 0.002}$ | $0.686_{\pm 0.056}$ | $0.348_{\pm 0.091}$ | $0.460_{\pm 0.149}$ | $0.792_{\pm 0.015}$ | $0.773_{\pm 0.020}$ |
| Citeseer | $0.822_{\pm 0.017}$ | $0.771_{\pm 0.020}$ | $0.707_{\pm 0.068}$ | $0.249_{\pm 0.168}$ | $0.472_{\pm 0.196}$ | $\underline{0.858}_{\pm 0.020}$ | $\mathbf{0.868}_{\pm 0.023}$ |
| Amazon-Photos | $0.642_{\pm 0.029}$ | $0.430_{\pm 0.032}$ | $0.486_{\pm 0.025}$ | $0.369_{\pm 0.013}$ | $0.434_{\pm 0.038}$ | $0.562_{\pm 0.013}$ | $0.517_{\pm 0.016}$ |
| Amazon-Computers | $0.426_{\pm 0.036}$ | $0.320_{\pm 0.060}$ | $0.240_{\pm 0.027}$ | $0.201_{\pm 0.032}$ | $0.258_{\pm 0.008}$ | $0.346_{\pm 0.018}$ | $0.315_{\pm 0.015}$ |
| Coauthor-Cs | $0.762_{\pm 0.010}$ | $0.787_{\pm 0.011}$ | $0.456_{\pm 0.066}$ | $0.229_{\pm 0.018}$ | $0.298_{\pm 0.033}$ | $0.515_{\pm 0.016}$ | $0.555_{\pm 0.009}$ |
| Coauthor-Physics | $0.798_{\pm 0.018}$ | $0.810_{\pm 0.003}$ | OOM | $0.157_{\pm 0.009}$ | $0.187_{\pm 0.011}$ | $0.476_{\pm 0.015}$ | $0.471_{\pm 0.021}$ |

## A.7 CORRUPTIONS

In this work, we experiment with the following corruptions:

1. RANDOMFEATRANDOMEDGE: Randomly generate an adjacency matrix $\tilde{A}$ and $\tilde{X}$ with the same sizes as $A$ and $X$, respectively. Note that $\tilde{A}$ and $A$ also have the same number of non-zero entries, i.e., the same number of edges.

2. SHUFFLEFEATRANDOMEDGE: Randomly shuffle the rows of $X$, and generate a random $\tilde{A}$ with the same size as $A$. Note that $\tilde{A}$ and $A$ also have the same number of non-zero entries, i.e., the same number of edges.

3. SPARSIFYFEATSPARSIFYEDGE: Mask out a large percentage (we chose 95%) of the entries in $X$ and $A$.

Of these corruptions, we find that SHUFFLEFEATRANDOMEDGE works the best across our experiments.

## A.8 AUC-ROC RESULTS

Here we include the area under the ROC curve for each of the different models under both the inductive and transductive settings. Note that we perform early stopping on the validation Hits@50 when training the link prediction model, not on the validation AUC-ROC.

Table 7: AUC-ROC of various methods in the inductive setting. See Section 3.1.2 for an explanation of our inductive setting.

| Dataset | End-To-End | Contrastive | | Non-Contrastive | | | |
| | E2E-GCN | ML-GCN | GRACE | GBT | CCA-SSG | BGRL | T-BGRL |
|---|---|---|---|---|---|---|---|
| | | | Overall | | | | |
| Cora | $0.788_{\pm0.015}$ | $0.842_{\pm0.008}$ | $\underline{0.858}_{\pm0.012}$ | $0.704_{\pm0.032}$ | $0.595_{\pm0.035}$ | $0.814_{\pm0.022}$ | $\mathbf{0.920}_{\pm0.008}$ |
| Citeseer | $0.810_{\pm0.016}$ | $0.873_{\pm0.004}$ | $0.886_{\pm0.010}$ | $0.691_{\pm0.007}$ | $0.621_{\pm0.070}$ | $\underline{0.891}_{\pm0.006}$ | $\mathbf{0.954}_{\pm0.003}$ |
| Coauthor-Cs | $0.881_{\pm0.040}$ | $0.956_{\pm0.001}$ | $0.944_{\pm0.001}$ | $0.875_{\pm0.036}$ | $0.831_{\pm0.068}$ | $\mathbf{0.968}_{\pm0.001}$ | $\underline{0.958}_{\pm0.001}$ |
| Coauthor-Physics | $0.957_{\pm0.004}$ | $\mathbf{0.976}_{\pm0.001}$ | OOM | $0.818_{\pm0.092}$ | $0.614_{\pm0.050}$ | $0.974_{\pm0.001}$ | $0.976_{\pm0.001}$ |
| Amazon-Computers | $0.974_{\pm0.009}$ | $\underline{0.981}_{\pm0.001}$ | $0.972_{\pm0.012}$ | $0.919_{\pm0.023}$ | $0.910_{\pm0.031}$ | $0.980_{\pm0.002}$ | $\mathbf{0.982}_{\pm0.002}$ |
| Amazon-Photos | $0.976_{\pm0.003}$ | $\underline{0.982}_{\pm0.001}$ | $0.977_{\pm0.002}$ | $0.962_{\pm0.011}$ | $0.885_{\pm0.057}$ | $\mathbf{0.984}_{\pm0.000}$ | $0.981_{\pm0.001}$ |
| | | Performance on Observed-Observed Node Edges | | | | | |
| Cora | $0.827_{\pm0.011}$ | $0.834_{\pm0.010}$ | $\underline{0.883}_{\pm0.010}$ | $0.714_{\pm0.034}$ | $0.584_{\pm0.047}$ | $0.800_{\pm0.025}$ | $\mathbf{0.929}_{\pm0.005}$ |
| Citeseer | $0.792_{\pm0.014}$ | $0.840_{\pm0.013}$ | $\underline{0.905}_{\pm0.012}$ | $0.705_{\pm0.019}$ | $0.635_{\pm0.078}$ | $0.875_{\pm0.006}$ | $\mathbf{0.956}_{\pm0.005}$ |
| Coauthor-Cs | $0.886_{\pm0.037}$ | $0.951_{\pm0.001}$ | $0.947_{\pm0.002}$ | $0.874_{\pm0.037}$ | $0.828_{\pm0.070}$ | $\mathbf{0.967}_{\pm0.001}$ | $\underline{0.955}_{\pm0.002}$ |
| Coauthor-Physics | $0.959_{\pm0.004}$ | $\mathbf{0.976}_{\pm0.001}$ | OOM | $0.819_{\pm0.091}$ | $0.615_{\pm0.049}$ | $0.974_{\pm0.001}$ | $\underline{0.975}_{\pm0.001}$ |
| Amazon-Computers | $0.974_{\pm0.010}$ | $\mathbf{0.981}_{\pm0.001}$ | $0.971_{\pm0.012}$ | $0.918_{\pm0.024}$ | $0.910_{\pm0.030}$ | $\underline{0.979}_{\pm0.002}$ | $0.981_{\pm0.002}$ |
| Amazon-Photos | $0.976_{\pm0.003}$ | $\underline{0.981}_{\pm0.001}$ | $0.977_{\pm0.002}$ | $0.962_{\pm0.011}$ | $0.885_{\pm0.055}$ | $\mathbf{0.983}_{\pm0.000}$ | $\underline{0.981}_{\pm0.001}$ |
| | | Performance on Observed-Unobserved Node Edges | | | | | |
| Cora | $0.741_{\pm0.022}$ | $\underline{0.844}_{\pm0.010}$ | $0.840_{\pm0.017}$ | $0.696_{\pm0.030}$ | $0.602_{\pm0.024}$ | $0.818_{\pm0.023}$ | $\mathbf{0.912}_{\pm0.010}$ |
| Citeseer | $0.841_{\pm0.019}$ | $0.901_{\pm0.005}$ | $0.877_{\pm0.012}$ | $0.687_{\pm0.016}$ | $0.610_{\pm0.069}$ | $\underline{0.904}_{\pm0.006}$ | $\mathbf{0.955}_{\pm0.004}$ |
| Coauthor-Cs | $0.877_{\pm0.045}$ | $\underline{0.964}_{\pm0.001}$ | $0.940_{\pm0.001}$ | $0.876_{\pm0.036}$ | $0.836_{\pm0.067}$ | $\mathbf{0.969}_{\pm0.001}$ | $0.964_{\pm0.001}$ |
| Coauthor-Physics | $0.953_{\pm0.004}$ | $\underline{0.975}_{\pm0.001}$ | OOM | $0.817_{\pm0.093}$ | $0.612_{\pm0.049}$ | $0.975_{\pm0.001}$ | $\mathbf{0.976}_{\pm0.000}$ |
| Amazon-Computers | $0.974_{\pm0.009}$ | $\underline{0.981}_{\pm0.001}$ | $0.973_{\pm0.011}$ | $0.921_{\pm0.022}$ | $0.909_{\pm0.032}$ | $0.980_{\pm0.002}$ | $\mathbf{0.982}_{\pm0.002}$ |
| Amazon-Photos | $0.977_{\pm0.003}$ | $\underline{0.983}_{\pm0.001}$ | $0.977_{\pm0.002}$ | $0.963_{\pm0.012}$ | $0.884_{\pm0.059}$ | $\mathbf{0.986}_{\pm0.000}$ | $0.981_{\pm0.002}$ |
| | | Performance on Unobserved-Unobserved Node Edges | | | | | |
| Cora | $0.571_{\pm0.043}$ | $\underline{0.879}_{\pm0.016}$ | $0.810_{\pm0.019}$ | $0.693_{\pm0.039}$ | $0.626_{\pm0.040}$ | $0.866_{\pm0.024}$ | $\mathbf{0.911}_{\pm0.011}$ |
| Citeseer | $0.852_{\pm0.047}$ | $\underline{0.917}_{\pm0.021}$ | $0.827_{\pm0.029}$ | $0.637_{\pm0.052}$ | $0.599_{\pm0.062}$ | $0.916_{\pm0.012}$ | $\mathbf{0.941}_{\pm0.011}$ |
| Coauthor-Cs | $0.850_{\pm0.059}$ | $0.964_{\pm0.001}$ | $0.928_{\pm0.004}$ | $0.877_{\pm0.034}$ | $0.839_{\pm0.061}$ | $\mathbf{0.967}_{\pm0.002}$ | $\underline{0.966}_{\pm0.001}$ |
| Coauthor-Physics | $0.949_{\pm0.006}$ | $\underline{0.978}_{\pm0.001}$ | OOM | $0.818_{\pm0.091}$ | $0.613_{\pm0.056}$ | $0.978_{\pm0.001}$ | $\mathbf{0.981}_{\pm0.001}$ |
| Amazon-Computers | $0.970_{\pm0.010}$ | $\mathbf{0.979}_{\pm0.001}$ | $0.969_{\pm0.011}$ | $0.914_{\pm0.022}$ | $0.899_{\pm0.035}$ | $\underline{0.977}_{\pm0.002}$ | $0.979_{\pm0.003}$ |
| Amazon-Photos | $0.978_{\pm0.004}$ | $\underline{0.982}_{\pm0.002}$ | $0.977_{\pm0.002}$ | $0.965_{\pm0.011}$ | $0.886_{\pm0.063}$ | $\mathbf{0.986}_{\pm0.001}$ | $\underline{0.982}_{\pm0.003}$ |

## A.9  WHY DOES BGRL NOT COLLAPSE?

The loss function for BGRL (see Equation (2)) is 0 when $h_i^{(2)} = 0$ or $\tilde{z}_i = 0$. While theoretically possible, this is clearly undesirable behavior since this does not result in useful embeddings. We refer to this case as the model collapsing. It is not fully understood why non-contrastive models do not collapse, but there have been several reasons proposed in the image domain with both theoretical and empirical grounding. Chen & He (2021) showed that the SimSiam architecture requires both the predictor and the stop gradient. This has also been shown to be true for BGRL. Tian et al. (2021) claim that the eigenspace of predictor weights will align with the correlation matrix of the online network under the assumption of a one-layer linear encoder and a one-layer linear predictor. Wen & Li (2022) looked at the case of a two-layer non-linear encoder with output normalization and found that the predictor is often only useful during the learning process, and often converges to the identity function. We did not observe this behavior on BGRL—the predictor is usually significantly different from that of the identity function.

## A.10  HOW DOES BGRL PULL REPRESENTATIONS CLOSER TOGETHER?

Here we clarify the intuition behind BGRL pulling similar points together. To simplify this analysis, we assume that the predictor is the identity function, which Wen & Li (2022) found is true in the image representation learning setting. Although we have not observed this in the graph setting, this assumption greatly simplifies our analysis and we argue it is sufficient for understanding why BGRL works.

Suppose we have three nodes: an anchor node $u$, a neighbor $v$, and a non-neighbor $w$. That is, we have $(u, v) \in \mathcal{E}$, $(u, w) \notin \mathcal{E}$, and $(v, w) \notin \mathcal{E}$. Let $\boldsymbol{u}, \boldsymbol{v}, \boldsymbol{w}$ be the embeddings for $u, v, w$, respectively (e.g. $\boldsymbol{u} = \text{ENC}(u)$).

Assuming homophily between the nodes, we have $\boldsymbol{u} \cdot \boldsymbol{v} < \boldsymbol{u} \cdot \boldsymbol{w}$. For ease of visualization, let us project the points in a 2D space. Then, we have the following:

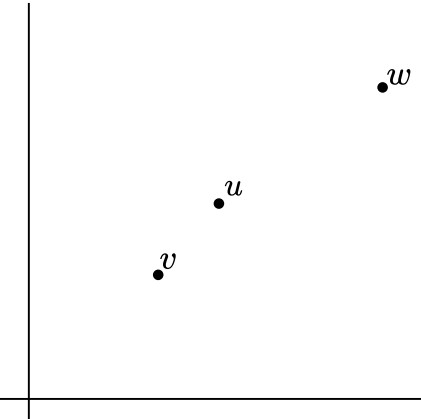

We then apply the two augmentations on $u$, producing $\tilde{u}_1 = \text{AUG}_1(u)$ and $\tilde{u}_2 = \text{AUG}_2(u)$. For the sake of simplicity, let us assume that we perform edge dropping and feature dropping with the same probability $p$ (in practice, they may be different from each other). We represent the space of possible values for $\tilde{u}_1$ and $\tilde{u}_2$ as a circle with radius $r$ centered at $u$, where $r$ is controlled by $p$.

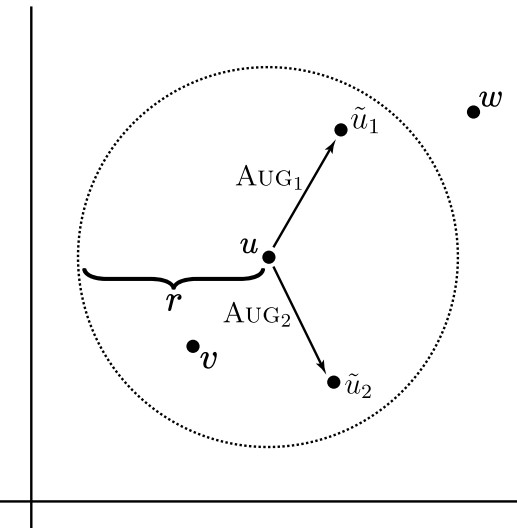

The BGRL loss is stated in Equation (2) above, but we rewrite it relative to our anchor $u$ and with our assumption about the predictor:

$$\mathcal{L}_u = -\frac{\tilde{\boldsymbol{u}}_1 \cdot \tilde{\boldsymbol{u}}_2}{||\tilde{\boldsymbol{u}}_1|| \, ||\tilde{\boldsymbol{u}}_2||} \tag{6}$$

Minimizing this loss pushes $\tilde{\boldsymbol{u}}_1$ and $\tilde{\boldsymbol{u}}_2$ closer. Let us denote the encoder after one round of optimization as $\text{ENC}'$. Then:

$$\mathbb{E}\left[||\text{ENC}'(\text{AUG}(u)) - \text{ENC}'(\text{AUG}(u))||\right] < \mathbb{E}\left[||\text{ENC}(\text{AUG}(u)) - \text{ENC}(\text{AUG}(u))||\right] \tag{7}$$

Note that $\boldsymbol{v}$ in this example lies within the space of possible augmentations - that is, $v \in \mathcal{A}$, where $\mathcal{A}$ is the set of all possible values of $\text{AUG}(u)$. This means, as we repeat this process, we implicitly push $\boldsymbol{u}$ and $\boldsymbol{v}$ closer together - leading to distributions like those shown in Figure 1.

## A.11 ADDITIONAL PLOTS

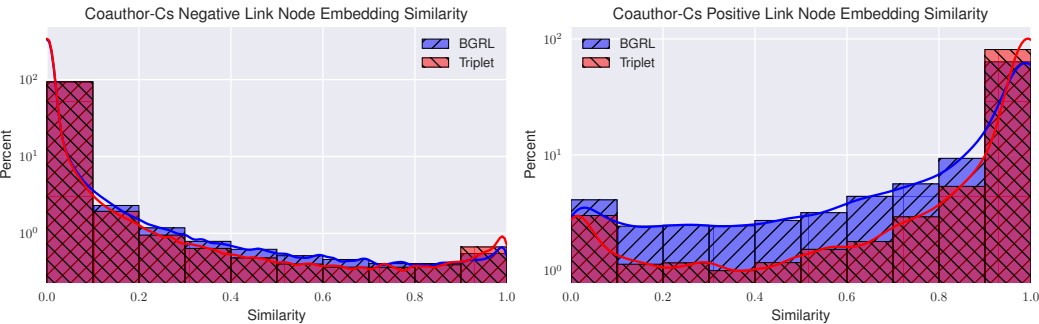

Figure 5: These plots show similarities between node embeddings on `Coauthor-Cs`. **Left:** distribution of similarity to *non-neighbors* for T-BGRL and BGRL (closer to 0 is better). **Right:** distribution of similarity to *neighbors* for T-BGRL and BGRL (closer to 1 is better). Note that the y-axis is on a logarithmic scale. T-BGRL clearly does a better job of ensuring that negative link representations are pushed far apart from those of positive links.

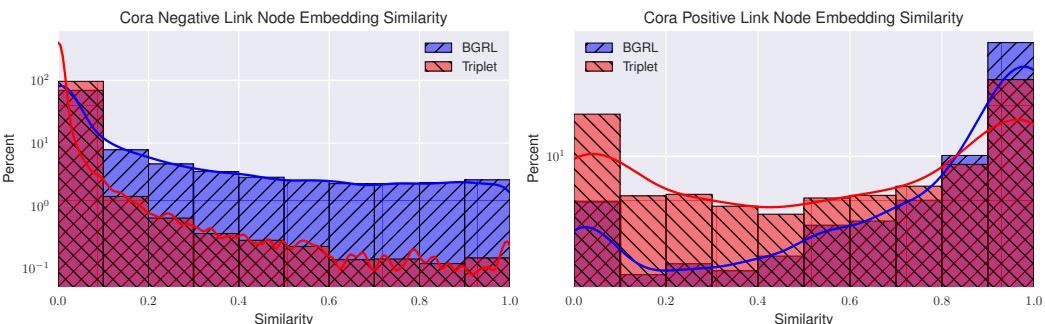

Figure 6: These plots show similarities between node embeddings on `Cora`. **Left:** distribution of similarity to *non-neighbors* for T-BGRL and BGRL (closer to 0 is better). **Right:** distribution of similarity to *neighbors* for T-BGRL and BGRL (closer to 1 is better). Note that the y-axis is on a logarithmic scale. T-BGRL clearly does a better job of ensuring that negative link representations are pushed far apart from those of positive links, but does not do as well at differentiating between positive links.

