# OpenReview forum: "Link Prediction with Non-Contrastive Learning"
_ICLR.cc/2023/Conference — ICLR 2023 poster_

### Official Review · Reviewer_N6ym · 2022-10-23

**Confidence:** 3
**Correctness:** 2
**Technical Novelty And Significance:** 2
**Empirical Novelty And Significance:** Not applicable
**Recommendation:** 5

**Clarity, Quality, Novelty And Reproducibility:**

The quality of this paper needs to be improved since it is not organised very well and some claims are not fully supported. The novelty is somewhat limited since the main content is focusing on the discussion of existing works while the proposed model is based on BGRL.

**Strength And Weaknesses:**

### Strength:

1. The discussion of contrastive and non-contrastive methods for the link prediction task is comprehensive.

### Weakness:

1. In Section 3.1, it’s unclear why the experiments only focus on the performance of the encoder and adapt the same decoder for all evaluated models. Moreover, it’s unclear why the encoder and decoder are not trained together. If this is following some existing works, please cite them and discuss the reason(s).
2. ML-GCN is only loosely defined but used as an important baseline. Is ML-GCN just a GCN model using the margin-loss? Also, what is ‘the supervised GCN’ (Section 3.1.1)? Is it just an end-to-end GCN?
3. Figure 1 does not have an x-axis.
4. ‘collapse’ is used without only once in Section 3.1.1 without definition and further discussion or claim.
5. Why ML-GCN and BGRL are both claimed as ‘our’ best-performing contrastive/non-contrastive models?
6. Is the SHUFFLEFEATRANDEDGE the same as the SHUFFLEFEATRANDOMEDGE method described in A.6? It’s unclear why a randomly generated adjacency matrix can help the effectiveness of BGRL. In fact, if the proposed method only generates a random matrix, then there is no corruption.
7. It’s unclear why some arguments/conclusions are stated in the caption of the figures e.g. figures 1&3.
8. It’s unclear why the early-stop is only used by ML-GCN.
9. This paper is poorly organised. Usually, there is no connection between two sections. Furthermore, many sections (e.g. the last subsection of Section 3.1.1) end without a solid conclusion or constructive advice.

### Typos:
1. On page 3, ‘none evaluate or target link prediction tasks.’  ’none evaluates or targets the link prediction task.’
2. On page 5, ‘which relative to an anchor node u is expressed as follows’. This sentence needs to be rewritten.
3. On page 5, ‘wen & Li (2022) looks at xxx’ ‘wen & Li (2022) looked at xxx’


**Summary Of The Paper:**

This paper first compares the performance of contrastive and non-contrastive methods on the graph link prediction task. Second, a novel model i.e. Triplet-BGRL (T-BGRL) is proposed based on Bootstrapped Graph Latents (BGRL). The proposed T-BGRL is categorised as a non-contrastive method and it relies on an efficient corruption method to generate cheap negative samples. Extensive experiments demonstrate that T-BGRL can achieve competitive performance for both the transductive and inductive settings with a good efficiency.

**Summary Of The Review:**

This paper needs to be polished, and some experimental setup needs to be justified to support the proposed claims.

---

> ### Author Response · Authors · 2022-11-15
> **Response to Review by Reviewer N6ym**
>
> Thank you for your detailed review. We would like to take the time to address each of your questions and concerns in detail below:
>
> ### 1) It is unclear why the experiments focus on the performance of the encoder and adapt the same decoder for all evaluated models. It’s also unclear why the encoder and decoder are not trained together.
>
> We fix the decoder model since we wish to focus on the choice of encoder for this paper. The choice of decoder has already been extensively studied [5,6], and we simply adopt the best-performing decoder (Hadamard product MLP). We train the encoder independently of the downstream task, following previous contrastive/non-contrastive works [2-4]. This also has the effect of maintaining the embeddings’ performance on other downstream tasks, like node classification (in which these models have already been shown to excel [2-4]). This is especially important for industrial and large-scale graph learning use cases, where we often have a multi-task embedding for each node stored in a feature store and multiple task-specific decoders. We have modified the paper to further clarify this point to future readers.
>
> ### 2) Is ML-GCN a GCN model using the margin loss? Also, what is the “supervised GCN”?
>
> Yes, the ML-GCN is a GCN model using the margin loss proposed in [1] (see Section 3.1.1 where we describe its max-margin loss function). The supervised GCN does refer to the end-to-end GCN, and we have corrected this typo.
>
> ### 3) Figure 1 does not have an x-axis
>
> Thank you for pointing this out. We have fixed this in the revised version and added an additional sentence to the caption to clarify this.
>
> ### 4) “collapse” is used once in Section 3.1.1 without further definition.
>
> We refer to collapse as the case where the model produces all-zero embeddings for every node, which would lead to a perfect loss of 0. We have clarified this point in the revised copy of the paper (revisions in blue). We have moved a more detailed discussion of the point to Appendix A.9 in the interest of space.
>
> ### 5) Why are ML-GCN and BGRL both claimed as ‘our’ best-performing contrastive/non-contrastive models?
>
> In that case, we meant to use the term “our” to refer to methods evaluated within this work. We recognize that this could be confusing and have removed it in the revised copy.
>
> ### 6) Is the SHUFFLEFEATRANDEDGE the same as the SHUFFLEFEATRANDOMEDGE method described in A.6? It’s unclear why a randomly generated adjacency matrix can help the effectiveness of BGRL.
>
> Yes, ShuffleFeatRandEdge and ShuffleFeatRandomEdge are the same. We have corrected this mismatch in the revised copy of the paper. The reason this corruption is useful is that it prevents the model from pushing all examples close together. We show this in Figure 3, where you can see that adding these examples pushes the positive/negative link distributions further apart. These corruptions have to be extreme in order to be effective - if they resemble possible real-world examples too closely, adding the corruptions may instead decrease performance. This is especially true in the inductive setting, where we will not know the features of new nodes and therefore want corrupted graphs to be very far away from the original.
>
> In the case of ShuffleFeatRandomEdge, we preserve the same number of edges in the randomized adjacency matrix. This means that we have the same average node degree, as well as the same features (just on different nodes). This can be viewed as edge removal followed by edge addition, with only a small number of original edges preserved. While this is aggressive, it does use some information from the original graph. We write the corruption as a random generation of the adjacency matrix because doing so is much faster than performing the two identical steps separately. We forgot to mention that we do preserve the same number of edges in the original paper and have added that detail to the revised edition.
>
> ### 7) It’s unclear why some arguments/conclusions are stated in the caption of the figures.
>
> All of the conclusions found in the captions can also be found in relevant sections in the text, but we include them in the figure for ease of reading and to distill relevant information for the reader. Please let us know if you feel that any of the particular arguments/conclusions in the captions are too strongly worded or misleading. If so, we will happily revise them.
>
> **We continue this response in the next comment due to length limitations**

---

> > ### Author Response · Authors · 2022-11-15
> > **Continuation of Response to Review by Reviewer N6ym**
> >
> > ### 8) It’s unclear why the early stopping is only used by ML-GCN.
> >
> > We do this for 3 reasons: (a) BGRL and other non-contrastive methods only compare a node to an augmented version of itself. These methods, therefore, do not directly make use of additional edges (outside of message-passing), unlike the ML-GCN, which can make direct use of additional edges in its loss function. The validation set only consists of withheld edges in both the transductive and inductive settings. As such, it is difficult to calculate a validation loss in the BGRL case to perform early stopping.  (b) The original BGRL paper [2] and implementation do not use early stopping and instead run for a fixed number of epochs. This is also true of the original GRACE [3] paper and implementation. (c) While we could conceivably perform early stopping on the training loss, we have found that it is continuously decreasing and is therefore not useful for early stopping.
> >
> > ### 9) There is usually no connection between two sections. Many sections (e.g. 3.1.1) end without a solid conclusion or constructive advice.
> >
> > We have attempted to resolve this issue in the revised copy of the paper. We have added a conclusion to Section 3.1.1, as well as improved the transitions between Sections 3.1.2–4, Sections 2–3, and Sections 3–3.1. We have also made various other changes to improve the clarity of the paper. All of these changes are shown in blue. Please let us know if you find that this is still an issue in the revised copy. We have also addressed all of the listed typos.
> >
> > ### Conclusion
> >
> > Thank you again for your thoughtful review of our paper. We hope that we have addressed each of your concerns and fixed the issues you mentioned in your review. Please let us know if this is not the case, and we will happily answer any additional questions you may have. If we have resolved each of the listed issues, we kindly ask you to consider increasing your rating.
> >
> > ### References
> > [1] Ying, Rex et al. “Graph convolutional neural networks for web-scale recommender systems” 24th ACM SIGKDD Conference on Knowledge Discovery and Data Mining. KDD, 2018
> >
> > [2] Thakoor, Shantanu, et al. “Large-Scale Representation Learning on Graphs via Bootstrapping” International Conference on Learning Representations. ICLR, 2022
> >
> > [3] Zhi, Yanqiao, et al. “Deep Graph Contrastive Representation Learning” ICML Workshop on Graph Representation Learning and Beyond. ICML-GRL+, 2020
> >
> > [4] Bielak, Piotr, et al. “Graph Barlow Twins: A self-supervised representation learning framework for graphs” Knowledge-Based Systems, 2022
> >
> > [5] Wang, Zhitao, et al. “Pairwise Learning for Neural Link Prediction”. arXiv, 2021
> >
> > [6] Wang, Yiwei, et al. “Flashlight : Scalable Link Prediction with Effective Decoders”. arXiv, 2022

---

> ### Author Response · Authors · 2022-11-29
> **Review Response Followup**
>
> Thank you again for your review! We have attempted to address your questions and concerns in our reply. If we have successfully done so, please consider raising your score as the end of the discussion period is approaching. If not, please let us know, and we will gladly address any additional questions/concerns you may have.

---

> ### Author Response · Authors · 2022-12-06
> **Reviewer N6ym Response Reminder**
>
> Thank you again for your review! As the end of the discussion period is near, we kindly remind you to take a look at our response and the updated manuscript (edits in blue) - which we believe has addressed all of your questions and concerns. If we have done so, please consider raising your score. If not, please let us know, and we will happily address any additional questions/concerns you may have.

---

> ### Author Response · Authors · 2022-12-08
> **Reminder for Reviewer N6ym**
>
> Dear Reviewer N6ym,
>
> Thank you for your review. We would like to remind you that the discussion period is drawing to a close (3 days left). We have previously sent a reply, which we believe addresses all of the points of concern in your review. We would appreciate it if you could take a look and let us know if that addresses all of your questions/concerns. If so, please consider raising your score. If not, please let us know, and we will gladly address any additional questions/concerns you may have.
>
> Thanks,
>
> Paper3389 Authors

---

### Official Review · Reviewer_myZB · 2022-10-24

**Confidence:** 4
**Correctness:** 4
**Technical Novelty And Significance:** 2
**Empirical Novelty And Significance:** 2
**Recommendation:** 6

**Clarity, Quality, Novelty And Reproducibility:**

I appreciate the effect of the authors to provide comprehensive experimental results of non-contrastive learning methods for link prediction, and also the intuitive analysis of their performance. However, this paper is more like a technical report of these non-contrastive learning methods to me, rather than a paper focusing on a new developed method.

1.The motivation is mainly derived from the experimental results rather than the theoretical analysis of shortcomings of existing non-contrastive learning methods for link prediction. In chapter 3.1.1, the author did not provide a convinced explanation for the reason why BGRL can work well in the transductive setting.

2.I can understand why the loss of ML-GCN can work well on link prediction, but what’s the connection between the loss of BGRL and ML-GCN? Why the loss of BGRL can also work well on link prediction?

3.The extension of BGRL is based on the hypothesis that ``One possible reason for the poor
performance of BGRL in the inductive setting is that it is unable to correctly differentiate unseen
positive from unseen negatives, i.e., it is overfitting on the training graph.’’ But there is no experiment to support this hypothesis.

4.Additionally, I am not sure whether other non-contrastive learning methods also perform poorly on link prediction. The best solution could be provide a theoretical analysis on why these methods tend to fail on link prediction.

5.Some suggestions.
It will be better to release the full name of BGRL when you first cite it.

6.Typo
Page 5. which relative to

**Strength And Weaknesses:**

Strength

1.The writing of this paper is good and easy to follow.
2.The experimental results on non-contrastive learning for link prediction is convinced and comprehensive.

Weakness
1.The novelty and theoretical analysis of the developed method is not enough

**Summary Of The Paper:**

This paper focuses on investigating the performance of existing
non-contrastive methods for link prediction in both transductive and inductive
settings. In their experiments, they find that BGRL generally performs well in transductive settings, but poorly in the more realistic inductive settings, which motivates them to develop an extension of BGRL named T-BGRL to alleviate its overfitting phenomenon.

**Summary Of The Review:**

This paper is more like a technical report of these non-contrastive learning methods to me, rather than a paper focusing on a new developed method.

---

> ### Author Response · Authors · 2022-11-15
> **Response to Review by Reviewer myZB**
>
> Thank you for your detailed review. We would like to address some of your questions/concerns below:
>
> We spend a portion of the paper analyzing the performance of existing non-contrastive methods, but we argue that this is a valuable and novel contribution, as we are the first work (to the best of our knowledge) to evaluate non-contrastive self-supervised learning for link prediction. This not only paves the way for future work in this direction but also allows us to find a key weakness in this class of methods (poor generalization to unseen examples, i.e., in the inductive setting). This then allows us to propose T-BGRL, a new approach based on BGRL, that improves its performance in the inductive setting. We would like to emphasize that the initial detailed evaluation of non-contrastive methods for link prediction is not only an important contribution to the body of literature but also what provides the motivation for our new method.
>
> ### 1) The author did not provide a convincing explanation for the reason why BGRL can work well in the transductive setting
>
> We added Appendix Section A.10 in the revised version of the paper, which provides a more detailed explanation of this point. It adds some intuition on why BGRL pulls the representations of neighboring nodes together, which leads to strong link prediction performance (see the paragraph titled “Ideal Link Prediction” in Section 3).
>
> ### 2) What’s the connection between the loss of BGRL and ML-GCN? Why can the loss of BGRL also work well on link prediction?
>
> We compare the behavior of BGRL and ML-GCN to show that, surprisingly, BGRL and ML-GCN behave similarly in that they both push together positives and (somewhat) push apart negatives. This can be seen in Figure 1, where we can see that node representations with links are pulled together and node representations without links remain further apart. We provide a detailed explanation of why this may be the case in Appendix Section A.10 of the revised paper.
>
> ### 3) There is no experiment to support the hypothesis that BGRL is unable to correctly differentiate unseen positives from unseen negatives.
>
> We empirically show this is indeed the case that BGRL is unable to correctly differentiate unseen positives from unseen negatives in Figure 3, where BGRL’s negative link distribution has heavy overlap with its positive link distribution. This plot is shown on the inductive examples and consists of edges to/from at least one unseen node. We have also found that this is true for the other datasets and is the basis for that statement. We have added additional figures for the Coauthor-Cs and Cora datasets in Appendix A.11. We have also modified the paper to further clarify this statement.
>
> ### 4) I am not sure whether other non-contrastive learning methods also perform poorly on link prediction. The best solution would be a theoretical analysis of why they fail.
>
> We show, empirically, that the other non-contrastive methods (we evaluate, to the best of our knowledge, all existing non-contrastive graph methods) perform generally worse than BGRL. However, we feel a detailed theoretical analysis of why the other methods fail is outside of the scope of this work. Our focus is instead on the empirical analysis of each method. Theoretically analyzing why even a single non-contrastive model works is already tricky, which is made more difficult due to the lack of existing analysis.
>
> For example, the original BGRL [1] and GBT [2] papers do not theoretically analyze why their methods work well for node classification, and the majority of the work is empirical. Even in the image case, recent works are just beginning to discover how exactly some non-contrastive models work [2] (which has 60+ pages of proofs). Furthermore, some of the works in the image area are still at odds with other. For example, [3] claims that BYOL only works with batch statistics, but its claims are refuted by [4]. [5] and [6] claim that the eigenspace of the predictor weights will align with the correlation matrix of the online network, but [2] finds that the predictor instead converges to the identity function. The SimSiam paper [7] claims that the predictor approximates the expectation over augmentations, but [8] claims that it does not.
>
> On top of this, no work (to the best of our knowledge) has transferred conclusions from non-contrastive image representation learning to non-contrastive graph learning (so we do not even know if the few results in the image domain transfer to the graph domain). As such, due to the scarcity of work in the area and conflicting conclusions, we rely largely on empirical results and leave a detailed theoretical analysis to future work.  That said, we do agree that theoretical analysis would be valuable, and we hope to explore these directions in future work.
>
> **We continue this response in the next comment due to length limitations**

---

> > ### Author Response · Authors · 2022-11-15
> > **Continuation of Response to Review by Reviewer myZB**
> >
> > ### 5) It will be better to release the full name of BGRL when you first cite it.
> >
> > Thank you, we have fixed this in the revised submission.
> >
> > ### 6) Typo on Page 5.
> >
> > Thank you, we have fixed this in the revised submission.
> >
> > ### Conclusion
> >
> > In addition to the above comments, we would like to emphasize the following points:
> >
> > - Our work is an exploratory work that finds that we can apply non-contrastive learning for link prediction. This is a valuable discovery that can greatly speed up existing link prediction systems. This can lead to the acceleration of recommendation systems and other large-scale industrial graph mining systems. With so few existing non-contrastive methods in graph learning, our work is one of the early works that provides further ground to flesh out the area.
> > - We show extremely strong empirical results: our proposed method, T-BGRL, is currently the best-performing non-contrastive method for link prediction, with much better performance than all of the other non-contrastive graph methods on the majority of datasets in realistic settings.
> > - T-BGRL’s performance is comparable to strong contrastive baselines but is up to 14x faster. Models similar to our baselines are used at large scale [9], and any speed increases could result in large real-world impacts.
> > - In response to Reviewer tANx, we have run experiments on a larger dataset (ogbl-collab) and shown that T-BGRL greatly improves over BGRL in the inductive setting.
> >
> > Thank you again for your thorough review of the paper. We have also made several modifications to the paper (shown in blue) to address your comments. Please let us know if anything is still unclear, and we will be happy to clarify! If we have successfully resolved all of your questions/concerns, we would appreciate it if you would consider raising your score.
> >
> > ### References
> >
> > [1] Thakoor, Shantanu, et al. “Large-Scale Representation Learning on Graphs via Bootstrapping” International Conference on Learning Representations. ICLR, 2022
> >
> > [2] Wen, Zixin, et al. “The Mechanism of Prediction Head in Non-contrastive Self-supervised Learning” arXiv, 2022
> >
> > [3] Fetterman, Abe and Albrecht, Josh. “'Understanding Self-Supervised and Contrastive Learning with "Bootstrap Your Own Latent' (BYOL)”. Online, 2020
> >
> > [4] Richemond, et al. “BYOL works even without batch statistics” arXiv, 2020
> >
> > [5] Tian, Yuandong, et al. “Understanding Self-Supervised Learning Dynamics without Contrastive Pairs”. Proceedings of the 38th International Conference on Machine Learning. PMLR, 2021
> >
> > [6] Wang, Xiang, et al. “Towards Demystifying Representation Learning with Non-contrastive Self-supervision”. OpenReview, 2022
> >
> > [7] Chen, Xinlei, et al. “Exploring Simple Siamese Representation Learning”. Computer Vision and Pattern Recognition. CVPR, 2020
> >
> > [8] Zhang, Chaoning, et al. “How Does SimSiam Avoid Collapse Without Negative Samples? A Unified Understanding with Self-supervised Contrastive Learning”. International Conference on Learning Representations. ICLR, 2022
> >
> > [9] Ying, Rex et al. “Graph convolutional neural networks for web-scale recommender systems” 24th ACM SIGKDD Conference on Knowledge Discovery and Data Mining. KDD, 2018

---

### Official Review · Reviewer_tANx · 2022-10-25

**Confidence:** 4
**Correctness:** 3
**Technical Novelty And Significance:** 3
**Empirical Novelty And Significance:** 3
**Recommendation:** 6

**Clarity, Quality, Novelty And Reproducibility:**

[1] This paper is clearly written and well organized.

[2] They have proof of the advantage of T-BGRL on small datasets but haven't evaluation on large datasets.

[3] This paper is an extension of BGRL and the implication of cheap “negative” samples is novel.

[4] The method is easy to reproducible.

**Strength And Weaknesses:**

Strength:

[1] This paper finds that BGRL tends to overfit the training graph, and thus can only perform well in the transductive setting. They introduce cheap “negative” samples to improve generalization on the indictive setting.

Weaknesses:

[1] They said it's the first work to explore link prediction with non-contrastive SSL methods. However, BGRL is a non-contrastive model and can be applied to link prediction tasks easily.

[2] These papers didn't compare with subgraph-based methods due to they are slow during inference. however, these subgraph-based methods perform well on large datasets. How about the performance of T-RGRL on large datasets (i.e. ogbl-ppa and ogbl-collab)?

**Summary Of The Paper:**

This paper follows BGRL, and notices that it generalizes poorly due to a lack of negative examples.
They propose a non-contrastive framework for link prediction named T-BGRL that uses cheap “negative” samples to improve generalization.
Experiments show that T-BGRL improves BGRL’s inductive performance in 5/6 datasets and is more efficient than contrastive methods.

**Summary Of The Review:**

The analysis of BGRL makes sense and the proposed T-BGRL performs well in the inductive setting.
I am open to raising my score if the authors can conduct experiments on ogbl-collab or ogbl-ppa.

---

> ### Author Response · Authors · 2022-11-15
> **Response to Review by Reviewer tANx**
>
> Thank you for your feedback! We would like to take the chance to respond to some of your comments:
>
> ### 1) BGRL is a non-contrastive model and can be applied to link prediction tasks easily.
>
> To the best of our knowledge, this work is indeed the first work to explore link prediction with non-contrastive SSL methods. We agree that it is not difficult to adapt BGRL for link prediction - although, to the best of our knowledge, it has not been done in existing work. More importantly, we also examine why it works (by showing that it unintuitively exhibits similar behavior to contrastive methods), find a key weakness in the method (doesn’t generalize well to unseen examples), and present a novel, efficient non-contrastive solution to this weakness without affecting its main advantage over contrastive methods (speed). We introduce T-BGRL, which makes use of corruption functions to greatly outperform BGRL (up to a 120% increase in Hits@50) in the inductive setting and achieve similar performance to the best contrastive baseline with a 14x speedup. Finally, we also provide a comprehensive set of benchmarks in both the transductive and inductive settings (see Tables 1 & 2) for non-contrastive self-supervised learning in link prediction.
>
> ### 2) Subgraph-based methods perform well on large datasets. How does T-BGRL perform on large datasets like ogbl-ppa or ogbl-collab?
>
> As you said, we do not evaluate the non-contrastive methods against subgraph-based methods due to subgraph-based methods having slow inference time - SUREL is ~250x slower, and SEAL is ~3900x slower than an end-to-end GCN [1]. It is true that these methods do perform much better, but it comes at the cost of greatly increased training and inference time - to the point where they are not feasible in real-world use cases like friend suggestion or large-scale recommendation systems. That is why the focus of our work is on more scalable node-embedding-based methods.
>
> Regarding larger datasets - that is a good point, and we have evaluated all of the baselines on ogbl-collab. The results are included in the revised copy of the paper. Please note that we used a single run as the best result (instead of the averaged results) due to computational resource limitations. We are currently running multiple runs for each of the experiments to obtain averaged numbers and will update the camera-ready (if accepted) with the averaged numbers and standard deviations.
>
> On collab, BGRL and T-BGRL are significantly outperformed by both the E2E-GCN and the ML-GCN. However, BGRL clearly exhibits the best performance among the non-contrastive methods and T-BGRL clearly improves the performance of BGRL in the inductive setting. The performance of BGRL greatly decreases in the inductive setting, but T-BGRL is able to counteract this decrease and has similar Hits@50 to BGRL in the transductive setting. The gap between T-BGRL and E2E-GCN also narrows in the inductive setting. These results show that T-BGRL is able to successfully improve the inductive performance of BGRL on large datasets.
>
> ### Conclusion
>
> I hope we have satisfactorily answered your questions. If so, please consider increasing our rating.  If not, please let us know, and we will be happy to answer any remaining questions!
>
> ### References
>
> [1] Yin, Haoteng, et al. “Algorithm and System Co-design for Efficient Subgraph-based Graph Representation Learning” VLDB, 2022

---

> > ### Comment · Reviewer_tANx · 2022-11-20
> > **After Rebuttal**
> >
> > Thanks for your reply, I will rise my score from 5 to 6.

---

> > > ### Author Response · Authors · 2022-11-22
> > > **Thank you for your update**
> > >
> > > Thank you for your help in strengthening the paper and for the reconsideration of your score! Please let us know if you have any further questions or concerns, and we would be more than happy to address them.

---

### Official Review · Reviewer_Qz7f · 2022-10-26

**Confidence:** 3
**Correctness:** 3
**Technical Novelty And Significance:** 3
**Empirical Novelty And Significance:** 2
**Recommendation:** 5

**Clarity, Quality, Novelty And Reproducibility:**

The paper is clear in terms of the algorithm and experiments but the importance and originality of the work needs to be addressed more clearly.

**Strength And Weaknesses:**

Strength:
1. The problem the paper is trying to solve is important research problem.
2. The method proposed is easy to implement and the experiments show that the performance did improve on most datasets.

Weakness:
1. I do not quite understand why we want to apply non-contrastive learning methods for link prediction. What is wrong with those contrastive learning methods?
2. Moreover, I do not understand why the proposed method falls in the category of non-contrastive learning methods. So it is very hard for me to judge the significance of the method.

**Summary Of The Paper:**

The paper is trying to tackle the area of graph self-supervised learning, which aims to derive useful node representations without labeled data. The paper proposes T-BGRL, a novel non-constrastive model to solve the link prediction problem.

**Summary Of The Review:**

Overall, the paper needs some improvement on explaining the significance of the work for it to be accepted.

---

> ### Author Response · Authors · 2022-11-15
> **Response to Review by Reviewer Qz7f**
>
> Thank you for your review! We would like to address some of the questions you had in your review.
>
> ### 1) Why do we want to apply non-contrastive methods for link prediction when we already have contrastive methods?
>
> We want to apply non-contrastive learning methods for link prediction to avoid the expensive negative sampling step required by existing graph contrastive methods (see Section 1). Figure 4  in our paper shows an example of this, where the best-performing contrastive method (ML-GCN) is roughly 14x slower on the coauthor-physics graph. This difference is further amplified on larger graphs, which is especially important for industrial and real-world use cases (where graphs can have billions of nodes and hundreds of billions of edges). We describe this in detail within the scalability paragraph of Section 4. Furthermore, contrastive methods tend to use much more memory than similar non-contrastive methods [1]. For example, BGRL uses 4x less memory than GRACE on Coauthor-Cs [1] and 8x less memory on Amazon-Computers. This can reduce training costs or further speed up training by allowing for larger minibatch sizes.
>
>
> ### 2) Why is the proposed method considered non-contrastive?
>
> We describe the following points in Section 4 (in the paragraph titled “Difference from Contrastive Methods”), but we elaborate here for clarity. The key difference between contrastive and non-contrastive learning is which samples we compare a given node to. Contrastive methods (see Definition 2.3) compare nodes to other samples in the dataset: positive (neighbors) and negative samples (non-neighbors). On the other hand, non-contrastive methods (see Definition 2.4) only compare nodes to variants of themselves (e.g., as a result of some augmentations). This may appear to be a subtle difference but can result in a significant speed-up and real-world impacts (as mentioned in point (1) above). If this is not fully convincing, the performance profile of T-BGRL is also very similar to other established non-contrastive methods like BGRL and GBT and very different from contrastive methods like GRACE or ML-GCN (which are 14x slower). This is shown in Figure 4 of the paper.
>
> ### 3) The importance and originality of the work need to be addressed more clearly.
>
> We would like to address your note on the importance and novelty of the work. We are (to the best of our knowledge) the first work to examine non-contrastive methods for link prediction, a very common and important problem that is applied to real-world use cases like recommendation and friend suggestion. We show that non-contrastive methods, with certain caveats, do indeed perform well for link prediction in the transductive setting (see Section 3.1.1), and present a novel method to address the cases where they do not, i.e., generalizing to unseen nodes (see Section 3.1.2). As a result, we provide detailed benchmarking experiments and explanations, in hopes that it will benefit future researchers interested in this topic. This provides a promising starting point for a whole direction of research that could significantly improve the efficiency of link prediction models.
>
> ### Conclusion
>
> We modified the manuscript to clarify the above points (with edits in blue), and we hope we have satisfactorily answered your questions. If so, please consider increasing your rating. If not, please let us know, and we will gladly answer any questions you may still have. Thank you!
>
> ### References
> [1] Thakoor, Shantanu, et al. “Large-Scale Representation Learning on Graphs via Bootstrapping” International Conference on Learning Representations. ICLR, 2022

---

> ### Author Response · Authors · 2022-11-29
> **Review Response Followup**
>
> Thank you again for your review! We have attempted to address your questions and concerns in our reply. If we have done so, please consider raising your score as the end of the discussion period is approaching. If not, please let us know, and we will gladly address any additional questions/concerns you may have.

---

> ### Author Response · Authors · 2022-12-06
> **Reviewer Qz7f Response Reminder**
>
> Thank you again for your review! As the discussion period is drawing to a close, please take a look at our reply. We believe we have answered all of your questions and concerns in our response and have added additional clarifying details in the modified manuscript (edits in blue). If we have not done so, please let us know, and we will gladly address any additional questions/concerns. Otherwise, please kindly consider raising your score.

---

> ### Author Response · Authors · 2022-12-08
> **Reminder for Reviewer Qz7f**
>
> Dear Reviewer Qz7f,
>
> Thank you for your review. We would like to remind you that the discussion period is drawing to a close (3 days left). We have previously sent a reply, which we believe addresses all of the points of concern in your review. We would appreciate it if you could take a look and let us know if that addresses all of your questions/concerns. If so, please consider raising your score. If not, please let us know, and we will gladly address any additional questions/concerns you may have.
>
> Thanks,
>
> Paper3389 Authors

---

### Author Response · Authors · 2022-11-15
**Rebuttal Revision to "Link Prediction with Non-Contrastive Learning"**

Thank you to all of the reviewers for your questions and comments. We have revised the manuscript to add additional clarifications and to fix any grammatical and structural mistakes. We have added preliminary experimental results for the OGB Collab dataset. We have also added Appendix A.10 (which provides some intuition on why BGRL pulls node neighbor representations together) and Appendix A.11 (which contains additional plots to support points made in the paper/rebuttal).

All of these changes have been made in blue to make it easier for readers to identify the changed portions. Please let us know if you have any questions or find any mistakes/typos in the revised text. Thank you!

---

### Decision · Program_Chairs · 2023-01-20

**Decision:**

Accept: poster

**Justification For Why Not Higher Score:**

The work is, in its present form, a borderline accept. Giving it a Spotlight would have required significantly stronger empirical and theoretical investigation.

**Justification For Why Not Lower Score:**

As discussed above, I have decided to recommend acceptance because of the complete lack of prior work in this area, the comprehensive empirical study conducted by the authors, and the potential immediate utility of the work, I am of the opinion the work in its current form will be quite interesting, and likely useful, to the ICLR audience.

**Metareview: Summary, Strengths And Weaknesses:**

The authors propose an empirical study for non-contrastive graph self-supervised learning (SSL) methods, in the context of link prediction. The arguments for this study are that:
* non-contrastive methods are significantly faster due to no need for generating negative samples, therefore making them more industrially relevant;
* link prediction tasks have significant downstream importance, but have been comparatively neglected by existing studies into non-contrastive SSL.

Therein, the authors first demonstrate that, despite opposite expectations, BGRL (a current state-of-the-art non-contrastive SSL method) appears to perform quite well in this regime, at least when the task is transductive. Qualitative analyses demonstrate one possible failure mode when moving into inductive tasks (overfitting to the training graph), and a modification, T-BGRL, is proposed. T-BGRL also includes a "negative" sample obtained by cheaply corrupting the target node's neighbourhood. The proposed T-BGRL method appears to significantly improve on BGRL on these tasks.

After the rebuttals and clarifications by the authors, the reviewers are split on whether to accept. The key remaining concerns appear to be:

* The lack of theoretical insight for the utility of (T-)BGRL in this context;
* The lack of motivation for the random corruption utilised in T-BGRL.

Having carefully analysed the paper and the reviewers' points, I have decided to discard both of these arguments and recommend (weak) acceptance. I fully agree that the paper would have benefitted from more rigorous analysis of these effects. However, especially given the complete lack of prior work in this area, the comprehensive empirical study conducted by the authors, and the potential immediate utility of the work, I am of the opinion the work in its current form will be quite interesting, and likely useful, to the ICLR audience. That being said I do hope that the authors will continue working on these problems in the future, and deepening the theoretical links!

Lastly, I was a bit surprised not to see VGAE [Kipf and Welling, NeurIPS BDL'16] cited or compared against when relevant. It's one of the simplest (and earliest) methods for self-supervised link prediction, albeit contrastive.

**Note From Pc:**

if the above contains the word "oral" or "spotlight" please see: "oral" presentation means -> notable-top-5% and "spotlight" means -> notable-top-25%. As stated in our emails, we are disassociating presentation type from AC recommendations